# A Unified Game-Theoretic Interpretation of Adversarial Robustness

**Jie Ren**[a*], **Die Zhang**[a*], **Yisen Wang**[b,c*], **Lu Chen**[a], **Zhanpeng Zhou**[a],
**Yiting Chen**[a], **Xu Cheng**[a], **Xin Wang**[a], **Meng Zhou**[a,d†], **Jie Shi**[e], **Quanshi Zhang**[a‡]

[a] Shanghai Jiao Tong University
[b] Key Lab. of Machine Perception, School of Artificial Intelligence, Peking University
[c] Institute for Artificial Intelligence, Peking University
[d] Carnegie Mellon University
[e] Huawei technologies Inc.

## Abstract

This paper provides a unified view to explain different adversarial attacks and defense methods, *i.e.* the view of multi-order interactions between input variables of DNNs. Based on the multi-order interaction, we discover that adversarial attacks mainly affect high-order interactions to fool the DNN. Furthermore, we find that the robustness of adversarially trained DNNs comes from category-specific low-order interactions. Our findings provide a potential method to unify adversarial perturbations and robustness, which can explain the existing defense methods in a principle way. Besides, our findings also make a revision of previous inaccurate understanding of the shape bias of adversarially learned features. Our code is available online at `https://github.com/Jie-Ren/A-Unified-Game-Theoretic-Interpretation-of-Adversarial-Robustness`.

## 1 Introduction

Adversarial robustness of deep neural networks (DNNs) has received increasing attention in recent years. Related studies include adversarial defense and attacks [48, 21]. In terms of defense, adversarial training is an effective and the most widely-used method [34, 72, 54, 55, 60]. In spite of their fast development, the essential mechanism of the adversarial robustness is still unclear. Thus, the understanding of adversarial attacks and defense is an emerging direction in recent years. Ilyas et al. [29] demonstrated adversarial examples could be attributed to the presence of non-robust yet discriminative features. Some methods [20, 58] explored the mathematical bound for the model robustness. Zhang and Zhu [73], Tsipras et al. [50] found adversarial training helped DNNs learn a more interpretable (more shape-biased) representation. Besides the feature interpretability, Tsipras et al. [50] further showed an inherent tension between the adversarial robustness and the generalization power.

Unlike above perspectives for explanations, we aim to propose a unified view to explain the essential reason why and how adversarial examples emerge, as well as essential mechanisms of various adversarial defense methods. We rethink the adversarial robustness from the novel perspective of

---

*Equal contribution

†This work was done when Meng Zhou was an undergraduate at Shanghai Jiao Tong University.

‡Quanshi Zhang is the corresponding author. `zqs1022@sjtu.edu.cn`. This work is supervised by Dr. Quanshi Zhang. He is with the John Hopcroft Center and the MoE Key Lab of Artificial Intelligence, AI Institute, at the Shanghai Jiao Tong University, China.

35th Conference on Neural Information Processing Systems (NeurIPS 2021).

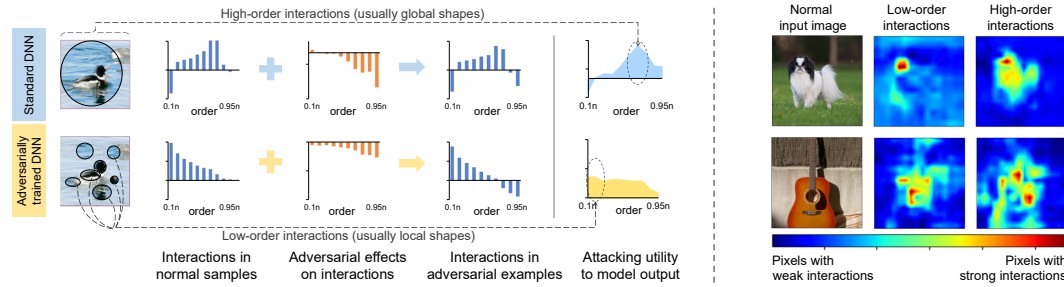

Figure 1: (Left) Adversarial attacks mainly affect high-order interactions in input samples. High-order interactions in adversarially trained DNNs are more robust than those in standard DNNs. (Right) Regions with strong low-order and high-order interactions, which are visualized by the method extended from [70].

interactions between input variables of a DNN. It is because in an adversarial example, adversarial perturbations on different pixels do not attack the DNN independently. Instead, perturbation pixels usually interact with each other to form a specific pattern for attacking. Surprisingly, we find that such interactions can explain various aspects of adversarial robustness. Specifically, this study aims to answer the following three questions.

• **How to disentangle feature representations that are sensitive to adversarial perturbations.** Based on the game-theoretic interactions, we aim to summarize the distinct property of feature representations, which are sensitive to adversarial perturbations, among overall feature representations.

• **How to explain the effectiveness of the adversarial training.** The above summarized property of sensitive feature representations also provides a new perspective to explain the utility of the adversarial training, *i.e.* why and how the adversarial training penalizes such sensitive feature representations.

• **How to unify various adversarial defense methods in a single theoretic system.** Our research provides a unified understanding for the success of adversarial defense methods [65, 15, 30].

As a prerequisite of analyzing the adversarial robustness, let us first revisit the interaction between input variables in DNNs. Let a set of input variables collaborate with each other to form an inference pattern, the Shapley interaction index [22] is a standard metric to measure the numerical benefits to the inference from their collaborations. This metric can be further extended to the multi-order interaction [70]. For the interaction between two input variables $(i, j)$, the interaction order measures the number of contextual variables that influence the significance of the interaction between $i$ and $j$. In other words, low-order interactions represent simple collaborations between input variables with small contexts, while high-order interactions indicate complex collaborations over large contexts (see Figure 1 (right)).

We further prove that the network output can be decomposed into the sum of multi-order interactions between different pairs of input variables. Thus, the overall effects of adversarial perturbations on the network output can be decomposed into elementary effects on different interaction components. Therefore, we can explain adversarial robustness using such elementary interaction components.

(1) **We discover and partially prove that adversarial perturbations mainly affect high-order interactions, rather than low-order interactions** (see Figure 1 (left)). In comparison, low-order interactions are naturally robust to attacks. In other words, adversarial attacks mainly affect the complex and large-scale collaborations among most pixels in the image. Based on this, we can successfully disentangle sensitive feature representations, *i.e.* high-order interactions.

*Interaction vs. frequency & rank.* Some studies explained adversarial examples as high-frequency features [66, 51, 23] and high-rank features [30]. We have conducted experiments to show that high-order interactions can better explain the essential property of attacking-sensitive representations, *i.e.* the complex and large-scale visual concepts.

(2) A clear difference between standard DNNs and adversarially trained DNNs is as follows. **Adversarial training significantly increases the robustness of high-order interactions.** In other words, attacks mainly affect complex collaborations in standard DNNs, while for adversarially trained DNNs, complex collaborations are not so vulnerable *w.r.t.* simple collaborations.

Then, we further explain the reason for the high robustness of adversarially trained DNNs. For interactions of each order, we define the disentanglement metric to identify whether interactions of this order are discriminative for the classification of a specific category, or represent common knowledge shared by different categories. For example, in Figure 1 (left), interactions representing the blue water may be shared by different categories, while interactions corresponding to the head of the red-breasted merganser are discriminative for this category. Based on the disentanglement metric, we discover that **compared with standard DNNs, adversarially trained DNNs usually encode more discriminative low-order interactions. Discriminative low-order interactions make high-order interactions of adversarially trained DNNs robust to attacks,** because contexts of high-order interactions are composed of many small contexts of low-order interactions. For example, if the simple (low-order) interactions for the red-breasted merganser are learned to be discriminative, instead of being shared by the bicycle category, then it is difficult to attack this image towards the bicycle category. It is because it is difficult to use the low-order interactions of the red-breasted merganser's head to construct high-order interactions of bicycles.

(3) **Our research provides a unified understanding for the success of several existing adversarial defense methods**, including the attribution-based detection of adversarial examples [65], the recoverability of adversarial examples to normal samples, the cutout method [15], and the rank-based method [30] (which is proved by [30] to be related to frequency-based methods [66, 51, 61]).

**Above findings also slightly revise the previous explanation of adversarially learned features** [21, 50, 16, 73]. They claimed that adversarial training learned more information about foreground shapes. We discover that these adversarially learned features are actually low-order interactions (usually local shapes), instead of modeling the global shape of the foreground.

**Explainable AI system based on game-theoretic interactions.** In fact, our research group led by Dr. Quanshi Zhang have proposed game-theoretic interactions, including interactions of different orders [69] and multivariate interactions [71]. The interaction can be used as an typical metric to explain signal processing in DNNs from different perspectives. For example, the game-theoretic metric can be used to guide the learning of baseline values of Shapley values [41]. Furthermore, we have built up a tree structure to explain the hierarchical interactions between words encoded by NLP models [68]. We have also used interactions to explain the generalization power of DNNs [70]. The interaction can also explain how adversarial perturbations contribute to the attacking task [53], and explain the transferability of adversarial perturbations [52]. Furthermore, we have also used the interaction to formulate the visual aesthetics [12] and signal-processing properties of different types of visual concepts [11] in DNNs. As an extension of the system of game-theoretic interactions, in this study, we explain the adversarial robustness based on interactions.

## 2 Related work

Attacking methods can be roughly summarized into white-box attacks [48, 21, 31, 38, 8] and black-box attacks [32, 39, 10, 6, 28, 59, 4]. For defense, adversarial training is one of the most effective and widely-used defense methods [34, 72, 54, 55, 60]. Other defense methods include masking gradients [39, 36], modifying networks [13, 19], and applying pre-processing on input images for testing [14, 35, 62, 3].

*Explanations for adversarial examples.* Some previous studies focused on the reason for the existence of adversarial examples. Goodfellow et al. [21] explained adversarial examples as a result of the high linearity of feature representations. Gilmer et al. [20], Ma et al. [33] proved that the existence of adversarial examples was due to the geometry of the high-dimensional manifold. Xie et al. [62], Xu et al. [63, 64], Bai et al. [5] discovered that adversarial perturbations usually activated substantial "noise" and semantically irrelevant features. Engstrom et al. [17] investigated the vulnerability of DNNs to rotations and translations. Tsipras et al. [50], Ilyas et al. [29] demonstrated that adversarial examples were attributed to non-robust yet discriminative features.

*Understandings of adversarial training.* Athalye et al. [2] suggested that adversarial training did not cause the obfuscated gradients phenomenon, which boosted the robustness. Goodfellow et al. [21], Dong et al. [16], Tsipras et al. [50], Zhang and Zhu [73] found that adversarially trained DNNs learned more shape-biased features than standard DNNs. Chalasani et al. [9] proved that adversarially trained DNNs *w.r.t.* the $\ell_\infty$ attack exhibited more sparse attributions. Tian et al. [49] showed that there is robustness imbalance among classes in adversarial training. Song et al. [46] considered the

adversarial training as the enumeration of all potential adversarial perturbations. Wang et al. [54] explained adversarial training from the perspective of min-max optimization. Yin et al. [66], Wang et al. [56], Harder et al. [23] discovered adversarial training pushed DNNs to utilize low-frequency components in inputs.

*Understanding of the robustness.* Szegedy et al. [48], Hein and Andriushchenko [25] computed Lipschitz constant to explain the robustness. Ignatiev et al. [27], Boopathy et al. [7] investigated the connection between network interpretability and adversarial robustness. Tsipras et al. [50] proved the inherent tension between adversarial robustness and standard generalization power. Fawzi et al. [18], Gilmer et al. [20], Weng et al. [58] proved lower/upper bounds on the robustness. Pal and Vidal [37] proposed a game-theoretic framework to understand the adversarial robustness, by formulating the game between attackers and defenders *w.r.t.* a trained DNN. Under the game-theoretic framework, Pal and Vidal [37] theoretically proved that the FGM attack and the random smoothing defense formed a Nash Equilibrium under some assumptions.

Unlike previous explanations of the existence of adversarial examples and adversarial robustness, we explain adversarial examples and adversarial training from a new perspective, *i.e.* the complexity of visual concepts that are learned by a DNN.

## 3   Decomposing attacking utility into interactions of multiple orders

**Preliminaries: using Shapley values [44] and the Shapley interaction index [22] to explain a DNN.** The Shapley value [44] in game theory is widely considered as an unbiased estimation for the importance or contribution of each player in a game. Given a trained DNN and the input with $n$ variables (*e.g.* an image with $n$ pixels, a sentence with $n$ words) $N = \{1, \cdots, n\}$, we can take input variables as players, and consider the network output as the reward. Shapley values can fairly divide and assign numerical effects on the network output to each input variable. More specifically, let $2^N \overset{\text{def}}{=} \{S | S \subseteq N\}$ denote all potential subsets of $N$, and each subset $S$ represents a specific context (*e.g.* a set of pixels in an image). $v(S) \in \mathbb{R}$ represents the scalar network output when we keep variables in $S$ unchanged and mask variables in $N \setminus S$ by following settings in [1] (*i.e.* setting a masked variable to the average value over different input samples).[4] In particular, $v(N)$ refers to the network output *w.r.t.* the entire input $N$ (*e.g.* the whole image), and $v(\emptyset)$ denotes the output when we mask all variables. In this way, the Shapley value $\phi(i)$ unbiasedly measures the importance of the variable $i$ to the network output.

$$\phi(i) = \sum_{S \subseteq N \setminus \{i\}} p(S) \left[ v(S \cup \{i\}) - v(S) \right], \quad p(S) \overset{\text{def}}{=} \frac{(n - |S| - 1)! |S|!}{n!} \tag{1}$$

In this way, the network output can be considered as the sum of elementary importances of input variables, *i.e.* $v(N) = \sum_{i \in N} \phi(i) + v(\emptyset)$. The Shapley value has been proved to satisfy four desirable properties, *i.e.* **linearity, nullity, symmetry** and **efficiency** properties, thereby being regarded as a fair method to allocate the network output to each input variable [57].

Input variables of a DNN do not contribute to the network output independently. Instead, different variables collaborate with each other to affect the network output, and we use the interaction to quantify the numerical utility of such collaborations. To this end, the Shapley interaction index [22] measures the utility of the collaboration between two input variables $(i, j)$ by examining **whether the absence/presence of an input variable $j$ will change the importance of the other variable $i$.** Their interaction is defined as $I(i, j) = \tilde{\phi}(i)_{j \text{ always present}} - \tilde{\phi}(i)_{j \text{ always absent}}$, where $\tilde{\phi}(i)_{j \text{ always present}}$ and $\tilde{\phi}(i)_{j \text{ always absent}}$ denote the importance of the variable $i$ when $j$ is always present and when $j$ is always absent, respectively (please see [22] for more details about $I(i, j)$).

**Decomposition of the existing Shapley interaction index into multiple orders.** Zhang et al. [70] further decomposes the Shapley interaction index $I(i, j)$ [22] into interactions of different orders.

$$I(i, j) = \frac{1}{n - 1} \sum_{m=0}^{n-2} I_{ij}^{(m)}, \quad I_{ij}^{(m)} = \mathbb{E}_{S \subseteq N \setminus \{i, j\}, |S| = m} \left[ \Delta v(i, j, S) \right] \tag{2}$$

where $\Delta v(i, j, S) = v(S \cup \{i, j\}) - v(S \cup \{i\}) - v(S \cup \{j\}) + v(S)$.

---

[4]For the DNN trained for multi-category classification, we set $v(S) = \log p(y = c |$ given an input with variables in $N \setminus S$ masked), where $c$ is selected as either the ground-truth category of the input, or the incorrectly classified category after attacks.

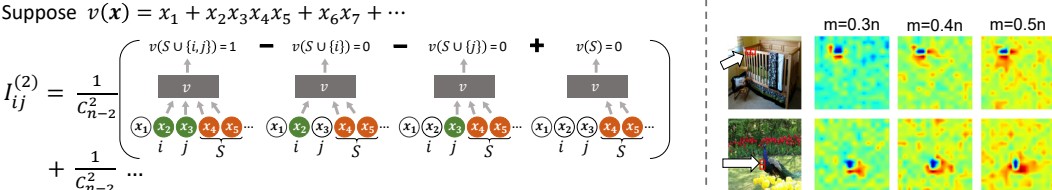

Figure 2: (Left) Computation of the $m$-order interaction between two input variables ($m = 2$). (Right) Contexts of the $m$-order interaction $I_{ij}^{(m)}$. Variables $(i, j)$ are indicated by red boxes. We extend the visualization method in [71], which visualized contexts of the Shapley interaction index, to show contexts of multi-order interactions (details in the supplementary material). Red (blue) colors indicate pixels that frequently (infrequently) appear in contexts which cause significant interactions.

**Proof of decomposing network output into multi-order interactions to explain adversarial attacks.** In this study, we successfully prove that the network output can be decomposed as the weighted sum of multi-order interactions. This enables us to use such interactions to explain adversarial attacks. Before the decomposition, let us first use interactions to understand the attack. As Figure 2 (left) shows, the interaction component of each order $I_{ij}^{(m)}$ represents collaborations between two input variables $(i, j)$ with a specific contextual complexity. Let us consider input variables $i, j$ and an arbitrary set of $m$ contextual variables. If $I_{ij}^{(m)} > 0$, it indicates that the presence of the variable $j$ will increase the importance of the variable $i$. Thus, we consider variables $(i, j)$ have a positive interaction. If $I_{ij}^{(m)} < 0$, it indicates a negative interaction.

The $m$-order interaction between two input variables $(i, j)$ measures the average interaction between $(i, j)$ when we consider their collaborations with $m$ contextual variables ($m$ pixels). From another perspective, the order $m$ measures the number of contextual variables that influence the interaction between $i$ and $j$. For a low order $m$, $I_{ij}^{(m)}$ reflects the interaction between $i$ and $j$ *w.r.t.* simple contexts of a few variables. For a high order $m$, $I_{ij}^{(m)}$ corresponds to the interaction *w.r.t.* complex contexts with massive variables. Figure 2 (right) visualizes the contexts $S$ corresponding to strong low-order interactions and strong high-order interactions, respectively. Cheng et al. [11] has also proven that low-order interactions (local collaborations) mainly reflect simple and common concepts (features), and high-order interactions (global collaborations) usually represent complex and global features.

• *Properties of multi-order interactions.* We have proven that $I_{ij}^{(m)}$ satisfies *linearity, nullity, commutativity, symmetry*, and *efficiency* properties (please see the supplementary material for details). In particular, the *efficiency property* is given as follows.

*Efficiency property:* The output of the DNN given the entire input $v(N)$ can be decomposed into interactions of different orders, *i.e.* $v(N) = v(\emptyset) + \sum_{i \in N} \phi^{(0)}(i) + \sum_{i,j \in N, i \neq j} \sum_{m=0}^{n-2} J_{ij}^{(m)}$, where $J_{ij}^{(m)} \stackrel{\text{def}}{=} \frac{n-1-m}{n(n-1)} I_{ij}^{(m)}$, and $\phi^{(0)}(i) \stackrel{\text{def}}{=} v(\{i\}) - v(\emptyset)$.

• *Decomposing the overall attacking utility into interactions of different orders.* According to the *efficiency property* of interactions, the output of the DNN *w.r.t.* the image $x$ can be decomposed into the sum of multi-order interactions between different pairs of input variables. Thus, we can decompose the overall utility of adversarial perturbations on the network output into elementary effects on interactions.

$$\Delta v(N|x) \stackrel{\text{def}}{=} v(N|x) - v(N|x^{\text{adv}}) = \underbrace{\Delta v(\emptyset|x) + \sum_{i \in N} \Delta \phi^{(0)}(i|x)}_{\text{usually can be ignored}} + \sum_{i \neq j \in N} \sum_{m=0}^{n-2} \Delta J_{ij}^{(m)}(x) \quad (3)$$

where $x \in \mathbb{R}^n$ denotes the normal sample, and $x^{\text{adv}} = x + \Delta x \in \mathbb{R}^n$ denotes the adversarial example. $v(N|x^{\text{adv}})$ denotes the network output given all variables in $N$ of the adversarial example $x^{\text{adv}}$. $v(N|x)$ corresponds to the normal sample. The first term $\Delta v(\emptyset) = v(\emptyset|x) - v(\emptyset|x^{\text{adv}}) = 0$. In the second term, $\Delta \phi^{(0)}(i|x) \stackrel{\text{def}}{=} (v(\{i\}|x) - v(\emptyset|x)) - (v(\{i\}|x^{\text{adv}}) - v(\emptyset|x^{\text{adv}})) = v(\{i\}|x) - v(\{i\}|x^{\text{adv}})$. Because in most applications, the importance of a single variable (*e.g.* a pixel) is usually small, we can ignore this term. In the third term, $\Delta J_{ij}^{(m)}(x)$ denotes the attacking utility of the $m$-order interaction between variables $(i, j)$ in $x$. $\Delta J_{ij}^{(m)}(x) \stackrel{\text{def}}{=} \frac{n-1-m}{n(n-1)} \Delta I_{ij}^{(m)}(x)$, where $\Delta I_{ij}^{(m)}(x) \stackrel{\text{def}}{=} I_{ij}^{(m)}(x) - I_{ij}^{(m)}(x^{\text{adv}})$ measures the elementary effects of adversarial perturbations on the $m$-order interaction.

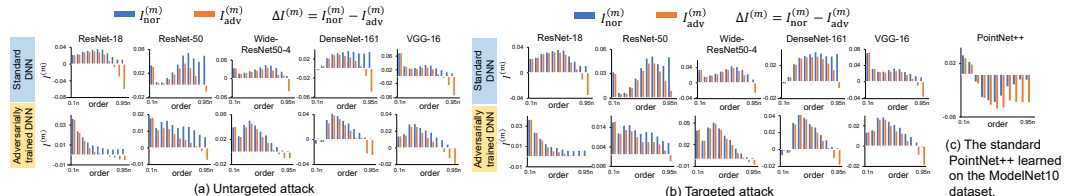

Figure 3: The multi-order interaction $I_{\text{nor}}^{(m)}$ and $I_{\text{adv}}^{(m)}$ of standard DNNs and adversarially trained DNNs. Adversarial perturbations mainly affected high-order interactions.

# 4  Explaining adversarial attacks and defense using interactions

To simplify the story, we only study the simplest and widely-used untargeted and targeted $\ell_\infty$ PGD attacks [34].

## 4.1  Attacks mainly affect high-order interactions

**Attacking utility of $m$-order interactions.** According to Eq. (3), effects of adversarial attacks mainly depend on changes of interactions $\Delta I_{ij}^{(m)}(x)$ (or $\Delta J_{ij}^{(m)}(x)$). Thus, in this section, we further conduct experiments to show that adversarial attacks affect $\Delta I_{ij}^{(m)}(x)$ of high orders. In order to compare the multi-order interactions in normal samples and those in adversarial examples, we measure the average interaction of a specific order $m$ among different pairs of variables in different input images, *i.e.* $I^{(m)} = \mathbb{E}_{x \in \Omega} \mathbb{E}_{i,j}[I_{ij}^{(m)}(x)]$, where $\Omega \subseteq \mathbb{R}^n$ denotes the set of all samples. We use the normal samples $\Omega_{\text{nor}}$ to compute $I_{\text{nor}}^{(m)}$ and use adversarial examples $\Omega_{\text{adv}}$ to compute $I_{\text{adv}}^{(m)}$. In this way, $\Delta I^{(m)} \overset{\text{def}}{=} I_{\text{nor}}^{(m)} - I_{\text{adv}}^{(m)}$ represents effects of attacks on the $m$-order interactions. Similarly, $\Delta J^{(m)} \overset{\text{def}}{=} \mathbb{E}_{x \in \Omega} \mathbb{E}_{i,j}[\Delta J_{ij}^{(m)}(x)] = \frac{n-1-m}{n(n-1)} \Delta I^{(m)}$ measures the attacking utility of the $m$-order interactions according to Eq. (3).

**Comparing attacking utilities between interactions of different orders (in terms of $I^{(m)}$).** In order to measure $I_{\text{nor}}^{(m)}$ and $I_{\text{adv}}^{(m)}$, we conducted experiments on ResNet-18/34/50 [24], Wide-ResNet50-4 [67], DenseNet-161 [26], and VGG-16 [45]. For each DNN, we obtained both the standardly trained version and the adversarially trained version on the ImageNet dataset [42][5]. We also conducted experiments on PointNet++ [40] learned on the ModelNet10 dataset [61], which is a 3D dataset. We measured the interaction $I^{(m)}$ by setting $v(S|x) = \log p(y = y^{\text{truth}}|$given variables in $S$ in the input $x$ and mask variables in $N \setminus S)$, and $x \in \Omega$ was sampled from the validation set. By following settings in [1], the masked variables in $N \setminus S$ were set to the average value over different input samples. However, the computational cost of $I^{(m)}$ was intolerable. To reduce the computational cost, we applied the sampling-based approximation method in [70] and computed interactions at the grid level, rather than the pixel level. Please see the supplementary material for more discussions.

We considered both untargeted and targeted[6] PGD attacks [34] with the $\ell_\infty$ constraint $\|\Delta x\|_{\ell_\infty} \le \epsilon$ to generate adversarial examples. We set $\epsilon = 32/255$ by following the setting in [62]. The step size was set to $2/255$. For fair comparisons, we controlled the perturbation generated for each image to have similar attacking utility of 8. The attacking utility in untargeted attacks was defined as $U_{\text{untarget}}(x) = h_{y^{\text{truth}}}(N|x) - h_{y^{\text{truth}}}(N|x^{\text{adv}})$, where $h_{y^{\text{truth}}}(N|x)$ and $h_{y^{\text{truth}}}(N|x^{\text{adv}})$ denote network outputs of the ground-truth category $y^{\text{truth}}$ before the softmax layer, when taking the normal sample and the adversarial example as input, respectively. The attacking utility in targeted attacks was defined as $U_{\text{target}}(x) = (h_{y^{\text{target}}}(N|x^{\text{adv}}) - h_{y^{\text{truth}}}(N|x^{\text{adv}})) - (h_{y^{\text{target}}}(N|x) - h_{y^{\text{truth}}}(N|x))$, where $h_{y^{\text{target}}}(\cdot)$ denote the network output of the target category[6] $y^{\text{target}}$.

*Results.* Figure 3 shows the multi-order interactions within normal samples and adversarial examples. We found that adversarial perturbations significantly decreased high-order interactions, in both adversarially trained DNNs and standard DNNs. For example, in Figure 3 (a), low-order interactions

---

[5]We used pretrained models released by Salman et al. [43].

[6]In the untargeted attack, we considered the misclassified category as the target category. In the targeted attack, the target label was set as the misclassified category in the untarget attack. When we consider interaction *w.r.t.* the target category, we set $v(S) = \log p(y = y^{\text{target}}|$given variables in $S$ and mask variables in $N \setminus S)$.

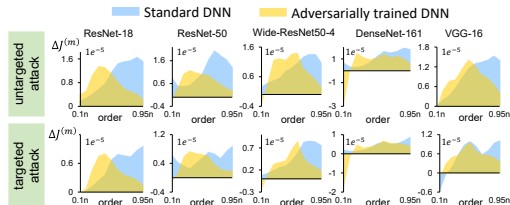

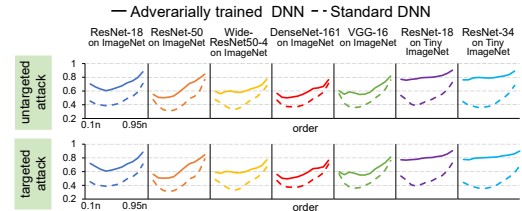

Figure 4: Distribution of compositional attacking utilities caused by interactions of different orders $\{\Delta J^{(m)}\}$.

Figure 5: The interaction disentanglement $D^{(m)}$. Interactions of adversarially trained DNNs exhibited higher disentanglement than those of standard DNNs.

$(m < 0.5n)$ in standard ResNet-18 decreased a little, while the interaction of order $m = 0.95n$ dropped from $0.01$ to $-0.059$. This phenomenon was consistent with the heuristic findings of Dong et al. [16] that neurons corresponding to high-level semantics were ambiguous.

*Additional experiments on the target category and other categories.* Beyond above analysis about interactions *w.r.t.* the ground-truth category, we also conducted experiments and found that high-order interactions *w.r.t.* the target category[6] increased significantly. Please see the supplementary material for details. In sum, high-order interactions were much more sensitive than low-order interactions, which verified our conclusions.

**Theoretic explanation of the sensitivity of high-order interactions.** Besides the above empirical observation, we can also theoretically prove that high-order interactions are more sensitive than low-order interactions.

***Proposition 1*** *(equivalence between the multi-order interaction and the mutual information): Given an input sample $x \in X \subseteq \mathbb{R}^n$ and the network output $Y$, we define $X_S = \{x_S | x \in X\}$ where $S \subseteq N$; each $x_S$ represents the sample, where variables not in $S$ are masked. If $v(S)$ is set as the entropy of classification $v(S) = H(Y|X_S) = \sum_{x_S} p(x_S) H(Y|X_S = x_S)$, then $I_{ij}^{(m)} = \mathbb{E}_{S \subseteq N \setminus \{i,j\}, |S|=m} MI(X_i; X_j; Y|X_S)$. The conditional mutual information $MI(X_i; X_j; Y|X_S)$ measures the remaining mutual information[7] between $X_i, X_j$ and $Y$, when $X_S$ is given. When $X_S$ (with $m$ pixels) is given, $MI(X_i; X_j; Y|X_S)$ can be considered as the benefit to the inference from the interaction between $X_i$ and $X_j$. Please see the supplementary material for proofs.*

Proposition 1 indicates that compared to low-order interactions, high-order interactions are conditioned on larger contexts $S$, *i.e.* conditioned on more contextual perturbations, thereby suffering more from adversarial perturbations.

Note that in Proposition 1, we set $\hat{v}(S) = H(Y|X_S)$, which is slightly different from setting $v(S) = \log p(y = y^{\text{truth}}|S, x)$ in Section 4.1. Nevertheless, the trend of $v(S)$ can roughly reflect the negative trend of $\hat{v}(S)$, which is discussed in the supplementary material.

**Comparison with frequency-based methods.** Previous studies [66, 51, 23] explained adversarial perturbations as high-frequency features. Figure 6 (left) shows the difference in the frequency between normal samples and adversarial samples. We find that high-order interactions can better distinguish adversarial examples and normal samples, than the frequency of features, which indicates that the interaction metric can better explain the essential property of adversarial perturbations. Please see the supplementary material for more details.

### 4.2 Explaining specific interactions encoded by adversarially trained DNNs

**Discovery.** In order to understand the difference in signal-processing behaviors between adversarially trained DNNs and standard DNNs, we have proven that the overall attacking utility $\mathbb{E}_{x \in \Omega}[\Delta v(N|x)]$ can be approximately decomposed as the sum of attacking utilities on multi-order interactions $\Delta J^{(m)}$, according to Eq. (3). Figure 4 shows the attacking utility of multi-order interactions *i.e.* $\Delta J^{(m)} = \frac{n-1-m}{n(n-1)} \Delta I^{(m)}$.

---

[7]Note that unlike the bivariate mutual information, $MI(X_i; X_j; Y|X_S)$ can be negative.

We noticed that for standard DNNs, adversarial attacks mainly affected high-order interactions. For adversarially trained DNNs, although high-order interactions were usually sensitive to attacks, low/middle-order interactions took up more attacking utilities. This phenomenon actually can also be observed in Figure 3, which is discussed in the supplementary material. Considering Eq. (3), this indicated that adversarial perturbations towards adversarially trained DNNs penalized both complex features of global collaborations and simple features of local collaborations.

**Explanation.** We further explain the above phenomena, *i.e.* why high-order interactions in adversarially trained DNNs are more robust to attacks than those in standard DNNs. To this end, we propose the following disentanglement metric for the interaction of a specific order.

$$D^{(m)} = \mathbb{E}_{x \in \Omega} \mathbb{E}_{\substack{i,j \in N \\ i \neq j}} \frac{|I_{ij}^{(m)}(x)|}{\mathbb{E}_{S \subseteq N \setminus \{i,j\}, |S|=m} |\Delta v(i,j,S|x)|} = \mathbb{E}_{x \in \Omega} \mathbb{E}_{\substack{i,j \in N \\ i \neq j}} \frac{|\mathbb{E}_{S \subseteq N \setminus \{i,j\}, |S|=m} \Delta v(i,j,S|x)|}{\mathbb{E}_{S \subseteq N \setminus \{i,j\}, |S|=m} |\Delta v(i,j,S|x)|} \quad (4)$$

The above disentanglement metric examines whether or not the $m$-order interactions represent discriminative information for a specific category. The high value of $D^{(m)}$ indicates that the $m$-order interactions relatively purely describe specific categories. In other words, the interactions between $(i,j)$ under different contexts $S$ consistently have the same effects (either positive or negative) on the inference of a specific category. Let us consider the following toy example. When the pair of $(i,j)$ consistently have positive interactions towards a specific category under different contexts, *i.e.* $\forall S \subseteq N \setminus \{i,j\}, |S| = m, \Delta v(i,j,S|x) > 0$, then we have $D^{(m)} = 1$. This means that the $m$-th interactions between $i$ and $j$ stably promote the output probability of this category. In contrast, a low value of $D^{(m)}$ indicates that interactions between $(i,j)$ represent diverse categories. *I.e.* given different contexts $S$, interactions between $(i,j)$ sometimes have positive effects on a specific category, and sometimes have negative effects. Please see the supplementary material for more discussions.

*Experiments.* Figure 5 compares the interaction disentanglement $D^{(m)}$ of standard DNNs and the disentanglement of adversarially trained DNNs. Interactions of adversarially trained DNNs were more disentangled than those of standard DNNs, especially for low-order interactions. This indicated that low-order interactions in adversarially trained DNNs encoded more category-specific information for inference than low-order interactions in standard DNNs.

Based on this observation, we could explain the robustness of high-order interactions to attacks in adversarially trained DNNs. Adversarial training learned more category-specific low-order interactions, which boosted the difficulty of attacking high-order interactions. This was because high-order interactions (usually global features) were usually constructed by low-order interactions (usually local features). Let us take the peacock in Figure 2 (right) for example. In this image, low-order interactions (simple features) represented the body of the peacock and the green garden. In this case, it was difficult to attack this image to other categories (*e.g.* the bicycle category) by constructing high-order interactions of the bicycle category using these peacock low-order features. Due to the difficulty of attacking high-order interactions, the adversarial examples towards the adversarially trained DNN had to attack low-order interactions, instead.

### 4.3 Unifying four existing adversarial defense methods

• **Explaining the attribution-based method of detecting adversarial examples.** Yang et al. [65] proposed a method to use the attribution score of input variables to detect adversarial examples. In order to prove the effectiveness of this method, we define the multi-order Shapley value just like the multi-order interaction, as follows. $\phi^{(m)}(i|x) \stackrel{\text{def}}{=} \mathbb{E}_{S \subseteq N \setminus \{i\}, |S|=m}[v(S \cup \{i\}|x) - v(S|x)]$, where $m$ denotes the order of the Shapley value. Then, we can prove that the attribution score which was used in [65] to detect adversarial exmaples, can be writtern as $\phi^{(n-1)}(i|x) = v(N|x) - v(N \setminus \{i\}|x)$. We further prove that $\phi^{(m)}(i|x)$ can be decomposed into interactions of different orders.

$$v(N|x) - v(\emptyset|x) = \frac{1}{n} \sum_{i \in N} \sum_{m=0}^{n-1} \phi^{(m)}(i|x), \quad \phi^{(m)}(i|x) = \mathbb{E}_{j \in N \setminus \{i\}} \left[ \sum_{k=0}^{m-1} I_{ij}^{(k)}(x) \right] + \phi^{(0)}(i|x) \quad (5)$$

According to the above equation, the overall adversarial effects can also be decomposed into elementary effects on $\phi^{(m)}(i|x)$. $\phi^{(0)}(i|x) = v(i|x) - v(\emptyset|x)$ is usually small and can be ignored, as discussed below Eq. (3). Among Shapley values of all orders, only the $(n-1)$-order component $\phi^{(n-1)}(i|x)$ contains the interactions of the highest order, which are the most sensitive interactions, according to Section 4.1. Therefore, the Shapley value component $\phi^{(n-1)}(i|x)$ is supposed to be

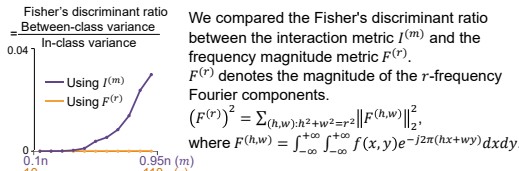
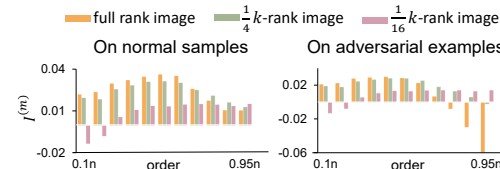

Figure 6: (Left) The game-theoretic interaction is a more significant metric to distinguish normal samples and adversarial examples. (Right) Multi-order interactions in images of different ranks.

Table 1: Ratio of adversarial examples whose classification results were corrected.

| Standard DNNs | cutout | ours | | | Adversarially trained DNNs | cutout | ours | | |
|---|---|---|---|---|---|---|---|---|---|
| | | $\alpha = 0.1$ | $\alpha = 0.2$ | $\alpha = 0.3$ | | | $\alpha = 0.1$ | $\alpha = 0.2$ | $\alpha = 0.3$ |
| ResNet-18 | 7.12% | 19.62% | 35.55% | **44.14%** | ResNet-18 | 23.92% | 22.46% | 33.78% | **39.18%** |
| ResNet-50 | 10.63% | 28.53% | 45.53% | **52.33%** | ResNet-50 | 24.27% | 23.23% | 35.23% | **41.54%** |
| DenseNet-161 | 11.32% | 27.16% | 46.15% | **56.35%** | DenseNet-161 | 35.65% | 24.65% | 39.25% | **45.74%** |

the most sensitive to adversarial perturbations, which proves the effectiveness of the detection of adversarial examples. Please see the supplementary material for related proofs and discussions.

• **Explaining high recoverability of adversarial examples towards adversarially trained DNNs.** The adversarial recoverability of a DNN is referred to as whether the DNN's adversarial examples can be inverted back to the normal sample by minimizing the classification loss. Specifically, given an adversarial example $x^{\text{adv}}$ generated by the untargeted attack [34] via $\max_{\|x^{\text{adv}}-x\|_p \leq \epsilon} \ell(x^{\text{adv}}, y^{\text{truth}})$, we conduct a targeted attack [34] on the adversarial sample to invert the classification result back to its ground-truth label, *i.e.* $\min_{\delta'} \ell(x^{\text{adv}} + \delta', y^{\text{truth}})$. Let $\hat{x} = x^{\text{adv}} + \delta'$ denote the recovered sample. If $\|\hat{x} - x\|_2 \leq \|x^{\text{adv}} - x\|_2$, it indicates a high recoverability; otherwise, a low recoverability.

We find that *adversarial examples generated by adversarially trained DNNs usually exhibit higher recoverability than those generated by standard DNNs.* This phenomenon can be explained by Proposition 1 in Section 4.1, *i.e.* low-order interactions correspond to the mutual information conditioned on smaller adversarial perturbations, thereby suffering less from attacks. Therefore, we can partially explain the high recoverability of adversarial examples towards adversarially trained DNNs. It is because adversarially trained DNNs mainly focus on low-order interactions (see Section 4.2), which leads to high recoverability.

Besides, we conducted experiments on ResNet-18/50 and DenseNet-161 trained on the ImageNet dataset, which also validated the high recoverability of adversarial examples generated on adversarially trained DNNs. Please see the supplementary material for details.

• **Explaining the rank-based method.** Jere et al. [30] discovered that standard DNNs paid certain attention to high-rank (*w.r.t.* the Singular Value Decomposition) features, but adversarial training put more attention on low-rank features. This implied that adversarial examples mainly contained high-rank features.

We used [30] to reduce the SVD rank of an image (either a normal sample or an adversarial example) to $\frac{1}{4}k$-rank image and $\frac{1}{16}k$-rank image, where $k$ denoted the full rank of the image. Figure 6 (right) shows that (1) compared with normal samples, the rank-reducing operation on adversarial examples significantly increased high-order interactions. This indicated our conclusions in Section 4.1 could explain that adversarial perturbations were mainly high-rank components in images. (2) When the rank of images was further reduced from $\frac{1}{4}k$ to $\frac{1}{16}k$, the rank-reducing operation affected both low-order and high-order interactions. However, when we reduce the rank of either normal samples or adversarial examples, they exhibited similar effects on low-order interactions, but exhibited dramatically different effects on high-order interactions. This indicated that **the most discriminative factor of distinguishing adversarial examples from normal samples was high-order interactions, rather than the rank-reducing operation.** In other words, the game-theoretic interaction presents a more essential property of adversarial attacks.

• **Explaining the cutout method.** DeVries and Taylor [15] proposed the cutout method to boost the robustness of DNNs by masking out a square region of the input during training. This operation can be roughly considered as applying a random dropout operation [47] to input images. The dropout operation on input images has been proven to destroy interactions between input variables [70]. To this end, we have further proven that the dropout operation mainly hurts high-order interactions, rather than low-order interactions (please see the proof in the supplementary material). In this way,

the cutout method can be considered as removing high-order interactions and pushing DNNs to learn more insensitive low-order interactions. Therefore, the learned DNN is robust to adversarial attacks.

In experiments, we applied the dropout operation to the misclassified adversarial examples, in order to verify the attacking-alleviating influence on the network output of removing high-order interactions. If the removal of high-order interactions could alleviate the effects of adversarial attacks, it would prove the attacking-alleviating influence of removing high-order interactions. Given ResNet-18/50 and DenseNet-161 trained on the ImageNet dataset, we selected normal samples that are correctly classified in the validation set of the ImageNet dataset, and generated adversarial examples with the $\ell_\infty$ untargeted PGD attack based on [62]. The attack was stopped once it succeeded. We applied the dropout operation with different dropout rates $\alpha$ at the pixel level, and the dropped pixels were filled with the average value of surrounding pixels. For the cutout method, we set the length of the side of the masked square regions as $112$, which was half of the side length of the input sample (dropping $\alpha = 25\%$ pixels of the input), following settings in [15]. Table 1 reports the ratio of adversarial examples whose classification results were corrected. When we applied dropout to adversarial examples, the classification accuracy of adversarial examples significantly increased, and even performed better than the cutout method. This indicated that the removal of high-order interactions presented a more essential defense mechanism than the cutout method.

## 5   Conclusion

In this paper, we have used the multi-order interaction to provide a unified understanding for the success of adversarial attacks and defense. Based on the multi-order interaction, we have explained adversarial attacking as mainly affecting high-order interactions between input variables. Furthermore, we have also explained the success of adversarial training, *i.e.* learning more category-specific low-order interactions to boost the robustness. Besides, we have further provided a unified explanation for some existing adversarial defense methods.

## Acknowledgments and Disclosure of Funding

This work is partially supported by National Science and Technology Innovation 2030 Major Project of the Ministry of Science and Technology of China under Grant (2021ZD0111602), the National Nature Science Foundation of China (No. 61906120, U19B2043), Shanghai Natural Science Foundation (21JC1403800,21ZR1434600), Shanghai Municipal Science and Technology Major Project (2021SHZDZX0102). This work is also partially supported by Huawei Technologies Inc. Prof. Yisen Wang is partially supported by the National Natural Science Foundation of China under Grant 62006153, and Project 2020BD006 supported by PKU-Baidu Fund. Xin Wang is supported by Wu Wen Jun Honorary Doctoral Scholarship, AI Institute, Shanghai Jiao Tong University.

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
