# A Unified Game-Theoretic Interpretation of Adversarial Robustness: Supplementary Material

**Jie Ren**[a*], **Die Zhang**[a*], **Yisen Wang**[b*], **Lu Chen**[a], **Zhanpeng Zhou**[a],
**Yiting Chen**[a], **Xu Cheng**[a], **Xin Wang**[a], **Meng Zhou**[a,c†], **Jie Shi**[d], **Quanshi Zhang**[a‡]

[a] Shanghai Jiao Tong University
[b]Key Lab. of Machine Perception (MoE), School of EECS, Peking University
[c] Carnegie Mellon University
[d] Huawei technologies Inc.

## A    Preliminaries: Shapley values

In this section, in order to help readers understand the metric in the paper, we first revisit the definition of the Shapley value [14], which is widely considered as an unbiased estimation of the numerical importance *w.r.t.* each input variable. In game theory, the complex system is usually represented as a game, where each input variable is taken as a player, and the output of this system is regarded as the total reward of all players. Given a game with multiple players (input variables) $N = \{1, 2, \cdots, n\}$, some players cooperate to pursue a high reward. Thus, the task is to divide the total reward, and fairly assign the divided elementary reward to each individual player. In this way, the elementary reward can be considered as the numerical importance of the corresponding variable to the complex system.

Let $2^N \stackrel{\text{def}}{=} \{S|S \subseteq N\}$ indicate all potential subsets of $N$. The game $v : 2^N \to \mathbb{R}$ is a function, which estimates the overall reward $v(S)$ earned by each specific subset of players $S \subseteq N$. In this way, the Shapley value, denoted by $\phi(i)$, represents the numerical importance of the player $i$ to the game $v$.

$$\phi(i) = \sum_{S \subseteq N \setminus \{i\}} \frac{(n - |S| - 1)!|S|!}{n!} \Big[ v(S \cup \{i\}) - v(S) \Big]. \tag{1}$$

Weber [22] has proven that the Shapley value is a unique method to fairly allocate overall reward to each player that satisfies following properties.

**(1) Linearity property**: If two independent games can be merged into one game $u(S) = v(S) + w(S)$, then the Shapley value of the new game also can be merged, *i.e.* $\forall i \in N, \phi_u(i) = \phi_v(i) + \phi_w(i); \forall c \in \mathbb{R}, \phi_{c \cdot u}(i) = c \cdot \phi_u(i)$.

**(2) Nullity property**: The dummy player $i$ is defined as a player satisfying $\forall S \subseteq N \setminus \{i\}, v(S \cup \{i\}) = v(S) + v(\{i\})$, which indicates that the player $i$ has no interactions with other players in $N$, $\phi(i) = v(\{i\}) - v(\emptyset)$.

**(3) Symmetry property**: If $\forall S \subseteq N \setminus \{i, j\}, v(S \cup \{i\}) = v(S \cup \{j\})$, then $\phi(i) = \phi(j)$.

**(4) Efficiency property**: The overall reward can be assigned to all players, $\sum_{i \in N} \phi(i) = v(N) - v(\emptyset)$.

**Using Shapley values to explain DNNs.** Given a trained DNN and the input with $n$ variables $N = \{1, \cdots, n\}$, we can take the input variables as players, and consider the network output as

---

[*]Equal contribution

[†]This work was done when Meng Zhou was an undergraduate at Shanghai Jiao Tong University.

[‡]Quanshi Zhang is the corresponding author. `zqs1022@sjtu.edu.cn`. This work is supervised by Dr. Quanshi Zhang. He is with the John Hopcroft Center and the MoE Key Lab of Artificial Intelligence, AI Institute, at the Shanghai Jiao Tong University, China.

35th Conference on Neural Information Processing Systems (NeurIPS 2021), Sydney, Australia.

the reward. The Shapley value $\phi(i)$ of each input variable $i \in N$ is regarded as the importance of the variable $i$ to the network output. Each subset of variables $S \subseteq N$ represents a specific context. $v(S)$ represents the network output, when we keep variables in $S$ unchanged and mask variables in $N \setminus S$ by following settings in [1]. In particular, $v(N)$ refers to the network output *w.r.t.* the entire input $N$, and $v(\emptyset)$ denotes the output when we mask all variables. Note that for the DNN trained for multi-category classification, $v(S)$ can be taken as an arbitrary dimension of the network output, so as to measure the variable importance to the corresponding category.

## B  Multi-order interactions and multi-order Shapley values

In this section, we first revisit the definition of the Shapley interaction index [6] and multi-order interactions [25], in order to improve the readability. Then, we provide proofs of extended properties of multi-order interactions, proofs of the relationship between multi-order interactions and multi-order Shapley values, and the proof of Proposition 1 in Section 4.1 of the paper.

### B.1  Multi-order interactions

**Shapley interaction index.**  Input variables of a DNN do not contribute to the network output independently. Instead, there are interactions/collaborations between different variables. To this end, the Shapley interaction index [6] measures the influence of a variable on the Shapley value (importance) of another variable. *I.e.* for two variables $(i, j)$, it examines whether the absence/presence of $j$ can change the importance of $i$. Thus, the Shapley interaction index is defined as the change in the Shapley value (importance) of variable $i$ when the variable $j$ is always present *w.r.t.* the case when $j$ is always absent, as follows.

$$I(i, j) = \tilde{\phi}(i)_{j \text{ always present}} - \tilde{\phi}(i)_{j \text{ always absent}} \tag{2}$$

where $\tilde{\phi}(i)_{j \text{ always present}}$ denotes the Shapley value of the variable $i$ computed under the specific condition that $j$ is always present. $\tilde{\phi}(i)_{j \text{ always absent}}$ is computed under the specific condition that $j$ is always absent. Note that $I(i, j) = I(j, i)$. If $I(i, j) > 0$, it indicates that the presence of the variable $j$ will increase the importance of the variable $i$. Thus, we consider variables $(i, j)$ have a positive interaction. If $I(i, j) < 0$, it indicates a negative interaction.

**Multi-order interactions.**  Zhang et al. [25] decomposed the Shapley interaction index into interactions of different orders, as follows.

$$I_{ij}^{(m)} = \mathbb{E}_{S \subseteq N \setminus \{i,j\}, |S|=m}[\Delta v(i, j, S)], \tag{3}$$

where $\Delta v(i, j, S) \stackrel{\text{def}}{=} v(S \cup \{i, j\}) - v(S \cup \{i\}) - v(S \cup \{j\}) + v(S)$. $I_{ij}^{(m)}$ denotes the interaction of the $m$-th order, which measures the average interaction between variables $(i, j)$ under all contexts consisting of $m$ variables (*e.g.* a visual context consisting of $m$ pixels). For a low order $m$, $I_{ij}^{(m)}$ reflects the interaction between $i$ and $j$ *w.r.t.* simple contextual collaborations with a few variables. For a high order $m$, $I_{ij}^{(m)}$ corresponds to the interaction *w.r.t.* complex contextual collaborations with massive variables.

**Proofs of new properties of multi-order interactions claimed in the paper.**  In Section 3 of the paper, we claim that $I_{ij}^{(m)}$ satisfies *linearity, nullity, commutativity, symmetry,* and *efficiency* properties. Proofs are given as follows.

**(1) Linearity property**: If we merge outputs of two DNNs, $u(S) = w(S) + v(S)$, then, $\forall i, j \in N$, the interaction $I_{ij,u}^{(m)}$ *w.r.t.* the new output $u$ can be decomposed into $I_{ij,u}^{(m)} = I_{ij,w}^{(m)} + I_{ij,v}^{(m)}$.

• *Proof*:

$$
\begin{aligned}
I_{ij,u}^{(m)} &= \mathbb{E}_{S \subseteq N \setminus \{i,j\}, |S|=m}[\Delta u(S, i, j)] \\
&= \mathbb{E}_{S \subseteq N \setminus \{i,j\}, |S|=m}[\Delta v(i, j, S) + \Delta w(S, i, j)] \\
&= I_{ij,v}^{(m)} + I_{ij,w}^{(m)}
\end{aligned}
$$

$\square$

**(2) Nullity property**: The dummy variable $i \in N$ satisfies $\forall S \subseteq N\backslash\{i\}, v(S\cup\{i\}) = v(S)+v(\{i\})$. It means that the variable $i$ has no interactions with other variables, *i.e.* $\forall j \in N, I_{ij}^{(m)} = 0$.

● *Proof*:

$$I_{ij}^{(m)} = \mathbb{E}_{S\subseteq N\backslash\{i,j\},|S|=m}[v(S \cup \{i,j\}) - v(S \cup \{i\}) - v(S \cup \{j\}) + v(S)]$$
$$= \mathbb{E}_{S\subseteq N\backslash\{i,j\},|S|=m}\big[v(S \cup \{j\} \cup \{i\}) - v(S \cup \{j\}) - \big(v(S \cup \{i\}) - v(S)\big)\big]$$
$$= \mathbb{E}_{S\subseteq N\backslash\{i,j\},|S|=m}[v(\{i\}) - v(\{i\})] = 0$$

$\square$

**(3) Commutativity property**: $\forall i, j \in N, I_{ij}^{(m)} = I_{ji}^{(m)}$.

● *Proof*:

$$I_{ij}^{(m)} = \mathbb{E}_{S\subseteq N\backslash\{i,j\},|S|=m}[v(S \cup \{i,j\}) - v(S \cup \{i\}) - v(S \cup \{j\}) + v(S)]$$
$$= \mathbb{E}_{S\subseteq N\backslash\{i,j\},|S|=m}[v(S \cup \{i,j\}) - v(S \cup \{j\}) - v(S \cup \{i\}) + v(S)]$$
$$= I_{ji}^{(m)}$$

$\square$

**(4) Symmetry property**: If input variables $i, j \in N$ have same cooperations with other variables $\forall S \subseteq N\backslash\{i,j\}, v(S \cup \{i\}) = v(S \cup \{j\})$, then they have same interactions, $\forall k \in N\backslash\{i,j\}$, $I_{ik}^{(m)} = I_{jk}^{(m)}$.

● *Proof*:

$$I_{ik}^{(m)} = \mathbb{E}_{\substack{S\subseteq N\backslash\{i,j\},\\|S|=m}}[\Delta v(i, k, S)]$$

$$= \frac{m!(n - 2 - m)!}{(n - 2)!} \sum_{\substack{S\subseteq N\backslash\{i,k\},\\|S|=m}} [\Delta v(i, k, S)]$$

$$= \frac{m!(n - 2 - m)!}{(n - 2)!} \left( \sum_{\substack{S\subseteq N\backslash\{i,j,k\},\\|S|=m-1}} [\Delta v(i, k, S \cup \{j\})] + \sum_{\substack{S\subseteq N\backslash\{i,j,k\},\\|S|=m}} [\Delta v(i, k, S)] \right)$$

$$= \frac{m!(n - 2 - m)!}{(n - 2)!} \left( \sum_{\substack{S\subseteq N\backslash\{i,j,k\},\\|S|=m-1}} [\Delta v(j, k, S \cup \{i\})] + \sum_{\substack{S\subseteq N\backslash\{i,j,k\},\\|S|=m}} [\Delta v(j, k, S)] \right)$$

$$= \mathbb{E}_{\substack{S\subseteq N\backslash\{i,j\},\\|S|=m}}[\Delta v(j, k, S)]$$

$$= I_{jk}^{(m)}$$

$\square$

**(5) Efficiency property**: The output of the DNN can be decomposed into interactions of different orders, $v(N) = v(\emptyset) + \sum_{i\in N} \phi^{(0)}(i) + \sum_{i\in N} \sum_{j\in N\backslash\{i\}}[\sum_{m=0}^{n-2} \frac{n-1-m}{n(n-1)} I_{ij}^{(m)}]$, where $\phi^{(0)}(i) \overset{\text{def}}{=} v(i) - v(\emptyset)$.

● *Proof*:

$$v(N) = v(\emptyset) + \frac{1}{n}\sum_{i\in N}\sum_{m=0}^{n-1} \phi^{(m)}(i)$$

$$= v(\emptyset) + \frac{1}{n}\sum_{i\in N} \phi^{(0)}(i) + \frac{1}{n}\sum_{i\in N}\sum_{m=1}^{n-1} \left[\mathbb{E}_{j\in N\backslash\{i\}}\big[\sum_{k=0}^{k=m-1} I_{ij}^{(k)}\big] + \phi^{(0)}(i)\right]$$

$$= v(\emptyset) + \sum_{i\in N} \phi^{(0)}(i) + \frac{1}{n(n-1)}\sum_{i\in N}\sum_{j\in N\backslash\{i\}} \big[\sum_{m=1}^{n-1}\sum_{k=0}^{k=m-1} I_{ij}^{(k)}\big]$$

$$= v(\emptyset) + \sum_{i\in N} \phi^{(0)}(i) + \sum_{i\in N}\sum_{j\in N\backslash\{i\}}[\sum_{m=0}^{n-2} \frac{n-1-m}{n(n-1)} I_{ij}^{(m)}]$$

$\square$

## B.2 Multi-order Shapley values

In Section 4.3 of the paper, we define multi-order Shapley values. In the supplementary material, this section provides more details about multi-order Shapley values to help readers understand, and also provides proofs of extended properties of multi-order Shapley values.

We decompose the Shapley value $\phi(i)$ into different orders, as follows.

$$\phi(i) = \frac{1}{n} \sum_{m=0}^{n-1} \phi^{(m)}(i), \tag{4}$$

$$\phi^{(m)}(i) = \mathbb{E}_{S \subseteq N \setminus \{i\}, |S|=m}[v(S \cup i) - v(S)], \tag{5}$$

where $\phi^{(m)}(i)$ denotes the Shapley value of the $m$-th order. It measures the importance of the input variable $i$ to the network output with contexts consisting of $m \in \{0, \ldots, n-1\}$ variables.

For a low order $m$, $\phi^{(m)}(i)$ denotes the importance of the variable $i$, when $i$ cooperates with a few contextual variables for inference. For a high order $m$, $\phi^{(m)}(i)$ describes the importance of the variable $i$, which cooperates with massive contextual variables. In particular, $\phi^{(0)}(i) = v(i) - v(\emptyset)$ represents the numerical importance of $i$ without taking into account any contexts.

In addition, we have proven that $\phi^{(m)}(i)$ satisfies following properties.

**(1) Linearity property**: If we merge the outputs of two DNNs $u(S) = w(S) + v(S)$, then the Shapley values of input variables also can be added, *i.e.* $\forall i \in N$, $\phi_u^{(m)}(i) = \phi_w^{(m)}(i) + \phi_v^{(m)}(i)$.

• *Proof*:

$$\begin{aligned}
\phi_u^{(m)}(i) &= \mathbb{E}_{S \subseteq N \setminus \{i\}, |S|=m}[u(S \cup \{i\}) - u(S)] \\
&= \mathbb{E}_{S \subseteq N \setminus \{i\}, |S|=m}[w(S \cup \{i\}) + v(S \cup \{i\}) - w(S) - v(S)] \\
&= \mathbb{E}_{S \subseteq N \setminus \{i\}, |S|=m}[w(S \cup \{i\}) - w(S)] + \mathbb{E}_{S \subseteq N \setminus \{i\}, |S|=m}[v(S \cup \{i\}) - v(S)] \\
&= \phi_w^{(m)}(i) + \phi_v^{(m)}(i)
\end{aligned}$$

$\square$

**(2) Nullity property**: An input variable $i \in N$ is considered as a dummy player if $\forall S \subseteq N \setminus \{i\}$, $v(S \cup \{i\}) = v(S) + v(\{i\})$. Thus, the variable $i$ has no interactions with other variables, *i.e.* $\phi^{(m)}(i) = v(\{i\})$.

• *Proof*:

$$\begin{aligned}
\phi^{(m)}(i) &= \mathbb{E}_{S \subseteq N \setminus \{i\}, |S|=m}[v(S \cup \{i\}) - v(S)] \\
&= \mathbb{E}_{S \subseteq N \setminus \{i\}, |S|=m}[v(\{i\})] = v(\{i\})
\end{aligned}$$

$\square$

**(3) Symmetry property**: Given two input variables $i, j \in N$, if these two variables have same cooperations with all other variables $\forall S \subseteq N \setminus \{i, j\}$, $v(S \cup \{i\}) = v(S \cup \{j\})$, then $\phi^{(m)}(i) = \phi^{(m)}(j)$.

- *Proof*:

$$\phi^{(m)}(i) = \mathbb{E}_{S \subseteq N \setminus \{i\}, |S|=m}[v(S \cup \{i\}) - v(S)]$$

$$= \frac{m!(n-1-m)!}{(n-1)!} \sum_{\substack{S \subseteq N \setminus \{i\}, \\ |S|=m}} [v(S \cup \{i\}) - v(S)]$$

$$= \frac{m!(n-1-m)!}{(n-1)!} \left( \sum_{\substack{S \subseteq N \setminus \{i,j\}, \\ |S|=m-1}} [v(S \cup \{i,j\}) - v(S,j)] + \sum_{\substack{S \subseteq N \setminus \{i,j\}, \\ |S|=m}} [v(S \cup \{i\}) - v(S)] \right)$$

$$= \frac{m!(n-1-m)!}{(n-1)!} \left( \sum_{\substack{S \subseteq N \setminus \{i,j\}, \\ |S|=m-1}} [v(S \cup \{i,j\}) - v(S,i)] + \sum_{\substack{S \subseteq N \setminus \{i,j\}, \\ |S|=m}} [v(S \cup \{j\}) - v(S)] \right)$$

$$= \mathbb{E}_{S \subseteq N \setminus \{j\}, |S|=m}[v(S \cup \{j\}) - v(S)]$$

$$= \phi^{(m)}(j)$$

$\square$

**(4) Efficiency property**: The overall reward can be assigned to all players, $\frac{1}{n} \sum_{i \in N} \sum_{m=0}^{n-1} \phi^{(m)}(i) = v(N) - v(\emptyset)$.

- *Proof*:

$$v(N) - v(\emptyset) = \sum_{i \in N} \phi(i) = \frac{1}{n} \sum_{i \in N} \sum_{m=0}^{n-1} \phi^{(m)}(i)$$

$\square$

### B.3 Relationship between multi-order Shapley values and multi-order interactions

This section provides proofs of the relationship between multi-order Shapley values and multi-order interactions. We have proven that multi-order Shapley values and multi-order interactions satisfy the following *marginal attribution* and *accumulation* properties.

**(1) Marginal attribution property**: The marginal attribution of the $(m+1)$-th order Shapley values beyond the $m$-th order is equal to the average interaction of the $m$-th order between $i$ and all other variables. $\forall i, j \in N, i \neq j$, $\phi^{(m+1)}(i) - \phi^{(m)}(i) = \mathbb{E}_{j \in N \setminus \{i\}}[I_{ij}^{(m)}]$.

- *Proof*:

$$\phi^{(m+1)}(i) - \phi^{(m)}(i) = \mathbb{E}_{\substack{S' \subseteq N \setminus \{i\} \\ |S'|=m+1}} \left[ v(S' \cup \{i\}) - v(S') \right] - \mathbb{E}_{\substack{S \subseteq N \setminus \{i\} \\ |S|=m}} \left[ v(S \cup \{i\}) - v(S) \right]$$

$$= \mathbb{E}_{\substack{S \subseteq N \setminus \{i\} \\ |S|=m}} \left[ \mathbb{E}_{j \in N \setminus (S \cup \{i\})} \left[ v(S \cup \{j\} \cup \{i\}) - v(S \cup \{j\}) \right] \right] - \mathbb{E}_{\substack{S \subseteq N \setminus \{i\} \\ |S|=m}} \left[ v(S \cup \{i\}) - v(S) \right]$$

$$= \mathbb{E}_{\substack{S \subseteq N \setminus \{i\} \\ |S|=m}} \left[ \mathbb{E}_{j \in N \setminus (S \cup \{i\})} \left[ v(S \cup \{j\} \cup \{i\}) - v(S \cup \{j\}) - v(S \cup \{i\}) + v(S) \right] \right]$$

$$= \mathbb{E}_{\substack{S \subseteq N \setminus \{i\} \\ |S|=m}} \mathbb{E}_{j \in N \setminus (S \cup \{i\})} \left[ \Delta v(i, j, S) \right]$$

$$= \mathbb{E}_{j \in N \setminus \{i\}} \mathbb{E}_{\substack{S \subseteq N \setminus \{i,j\} \\ |S|=m}} \left[ \Delta v(i, j, S) \right]$$

$$= \mathbb{E}_{j \in N \setminus \{i\}}[I_{ij}^{(m)}]$$

$\square$

**(2) Accumulation property**: The $m$-th ($m > 0$) order Shapley value of the variable $i \in N$ can be decomposed into interactions of lower orders, $\phi^{(m)}(i) = \mathbb{E}_{j \in N \setminus \{i\}}[\sum_{k=0}^{m-1} I_{ij}^{(k)}] + \phi^{(0)}(i)$.

- *Proof*:

$$\phi^{(m)}(i) = \phi^{(m)}(i) - \phi^{(m-1)}(i) + \phi^{(m-1)}(i) - \phi^{(m-2)}(i) + \cdots - \phi^{(0)}(i) + \phi^{(0)}(i)$$

$$= \mathbb{E}_{j \in N \setminus \{i\}}[I_{ij}^{(m-1)}] + \mathbb{E}_{j \in N \setminus \{i\}}[I_{ij}^{(m-2)}] + \cdots + \mathbb{E}_{j \in N \setminus \{i\}}[I_{ij}^{(0)}] + \phi^{(0)}(i)$$

$$= \mathbb{E}_{j \in N \setminus \{i\}} \left[ \sum_{k=0}^{k=m-1} I_{ij}^{(k)} \right] + \phi^{(0)}(i)$$

$\square$

### B.4 Equivalence between the multi-order interaction and the mutual information

Proposition 1 in Section 4.1 of the paper shows the equivalence between the multi-order interaction and the mutual information. In the supplementary material, this section provides the proof of this proposition.

When a DNN outputs a probability distribution, we prove that the interaction between input variables can be represented in the form of mutual information. Without loss of generality, let us take the image classification task for example. Let $x \in X \subseteq \mathbb{R}^n$ denote an input image of the DNN. $x_i$ denotes the $i$-th pixel, and $X_i = \{x_i\}$. $\forall S \subseteq N$, we define $X_S = \{x_S | x \in X\}$; each $x_S$ represents the image, where pixels in $S$ remain unchanged, and other pixels $j \in N \setminus S$ are masked following settings of [1]. Let $y \in Y = \{y^1, \cdots, y^C\}$ denote the network prediction. In this way, given $x_S$ as the input, $p(y|x_S)$ denotes the output probability of the DNN. Let us set $v(S) = H(Y|X_S) = \sum_{x_S} p(x_S)H(Y|X_S = x_S)$, which measures the entropy of $y$ given the input $x_S$. Then we prove that

$$I_{ij}^{(m)} = \mathbb{E}_{S \subseteq N \setminus \{i,j\}, |S|=m} MI(X_i; X_j; Y|X_S) \tag{6}$$

● *Proof*:

$$
\begin{aligned}
I_{ij}^{(m)} &= \mathbb{E}_{S \subseteq N \setminus \{i,j\}, |S|=m} \Big[ v(S \cup \{i,j\}) - v(S \cup \{i\}) - v(S \cup \{j\}) + v(S) \Big] \\
&= \mathbb{E}_{S \subseteq N \setminus \{i,j\}, |S|=m} \Big[ - H(Y|X_S, X_i, X_j) + H(Y|X_S, X_i) + H(Y|X_S, X_j) - H(H|X_S) \Big] \\
&= \mathbb{E}_{S \subseteq N \setminus \{i,j\}, |S|=m} \Big[ H(Y|X_S, X_j) - H(Y|X_S, X_j, X_i) + H(Y|X_S, X_i) - H(Y|X_S) \Big] \\
&= \mathbb{E}_{S \subseteq N \setminus \{i,j\}, |S|=m} \Big[ MI(X_i; Y|X_S, X_j) - MI(X_i; Y|X_S) \Big] \\
&= \mathbb{E}_{S \subseteq N \setminus \{i,j\}, |S|=m} \Big[ MI(X_i; X_j; Y|X_S) \Big]
\end{aligned}
$$

$\square$

The conditional mutual information $MI(X_i; X_j; Y|X_S)$ measures the remaining mutual information between $X_i, X_j$ and $Y$ when $X_S$ is given. Note that unlike the bivariate mutual information, $MI(X_i; X_j; Y|X_S)$ can be negative. When $X_S$ (each $x_S \in X_S$ containing $m$ pixels) is given, we can roughly understand the conditional mutual information $MI(X_{\{i,j\}}; Y|X_S)$ as the additional benefits from $X_i$ and $X_j$ to classification. We prove that $MI(X_{\{i,j\}}; Y|X_S)$ can be decomposed into the exclusive benefits of $X_i$ (*i.e.* $MI(X_i; Y|X_j, X_S)$), the exclusive benefits of $X_j$ (*i.e.* $MI(X_j; Y|X_i, X_S)$), and the benefit shared by $X_i$ and $X_j$ (*i.e.* $MI(X_i; X_j; Y|X_S)$). Thus, $MI(X_i; X_j; Y|X_S)$ can be considered as the benefits from the interaction between $X_i$ and $X_j$.

$$MI(X_{\{i,j\}}; Y|X_S) = MI(X_i; Y|X_j, X_S) + MI(X_j; Y|X_i, X_S) + MI(X_i; X_j; Y|X_S) \tag{7}$$

- *Proof*:

$\text{right} = MI(X_i; Y|X_j, X_S) + MI(X_j; Y|X_i, X_S) + MI(X_i; X_j; Y|X_S)$

$\quad = MI(X_i; Y|X_j, X_S) + MI(X_j; Y|X_i, X_S) + MI(X_i; Y|X_S) - MI(X_i; Y|X_j, X_S)$

$\quad = MI(X_i; Y|X_j, X_S) + MI(X_i; Y|X_S)$

$\quad = \sum_{x_i, x_j, x_S, y} p(x_i, x_j, x_S, y) \log \frac{p(x_j, y|x_i, x_S)}{p(x_j|x_i, x_S)p(y|x_i, x_S)} + \sum_{x_i, x_S, y} p(x_i, x_S, y) \log \frac{p(x_i, y|x_S)}{p(x_i|x_S)p(y|x_S)}$

$\quad = \sum_{x_i, x_j, x_S, y} p(x_i, x_j, x_S, y) \log \frac{p(x_j, y|x_i, x_S)}{p(x_j|x_i, x_S)p(y|x_i, x_S)} + \sum_{x_i, x_j, x_S, y} p(x_i, x_j, x_S, y) \log \frac{p(x_i, y|x_S)}{p(x_i|x_S)p(y|x_S)}$

$\quad = \sum_{x_i, x_j, x_S, y} p(x_i, x_j, x_S, y) \log \frac{p(x_j, y|x_i, x_S)p(x_i, y|x_S)}{p(x_j|x_i, x_S)p(y|x_i, x_S)p(x_i|x_S)p(y|x_S)}$

$\quad = \sum_{x_i, x_j, x_S, y} p(x_i, x_j, x_S, y) \log \frac{p(x_j, y|x_i, x_S)p(x_i, y|x_S)p(x_i, x_S)p(x_S)}{p(x_j|x_i, x_S)p(y|x_i, x_S)p(x_i|x_S)p(y|x_S)p(x_i, x_S)p(x_S)}$

$\quad = \sum_{x_i, x_j, x_S, y} p(x_i, x_j, x_S, y) \log \frac{p(x_i, x_j, x_S, y)p(x_i, x_S, y)}{p(x_j|x_i, x_S)p(x_i, x_S, y)p(x_i|x_S)p(y|x_S)p(x_S)}$

$\quad = \sum_{x_i, x_j, x_S, y} p(x_i, x_j, x_S, y) \log \frac{p(x_i, x_j, x_S, y)}{p(x_j|x_i, x_S)p(x_i|x_S)p(y|x_S)p(x_S)}$

$\quad = \sum_{x_i, x_j, x_S, y} p(x_i, x_j, x_S, y) \log \frac{p(x_i, x_j, y|x_S)}{p(x_i, x_j|x_S)p(y|x_S)}$

$\quad = MI(X_{\{i,j\}}; Y|X_S) = \text{left}$

$\square$

## C  Related works about interactions

In Section 2 of the paper, we have discussed related works about understandings of adversarial attacks, defense, and robustness. Due to the page limit, we discuss related works about interactions in this section of the supplementary material. Unlike previous studies about interactions, we firstly use the interaction to explain adversarial perturbations and robustness, and provide a unified view to understand existing defense methods.

Interactions between input variables of a DNN have been widely investigated in recent years. In game theory, Grabisch and Roubens [6] and Lundberg et al. [11] proposed and used the Shapley interaction index based on Shapley values [14]. Covert et al. [3] investigated the relationship between the Shapley value and the mutual information. Sorokina et al. [16] measured the interaction of multiple input variables in an additive model. Tsang et al. [19] calculated interactions of weights in a DNN. Wang et al. [21] applied the interaction of adversarial perturbations to understand adversarial transferabilitiy. Murdoch et al. [13], Singh et al. [15], and Jin et al. [10] used the contextual decomposition (CD) technique to extract variable interactions. Cui et al. [4] proposed a non-parametric probabilistic method to measure interactions using a Bayesian neural network. Janizek et al. [9] extended the Integrated Gradients method [17] to explain pairwise feature interactions in DNNs. Sundararajan et al. [18] defined the Shapley-Taylor index to measure interactions over binary features. In comparison, we novelly use the multi-order interaction to understand the detailed interaction behaviors *w.r.t.* adversarial attacks, which enables us to explain adversarial examples and adversarial training.

## D  Information reflected by $\Delta I^{(m)}$

In section 4.1 of the paper, we propose the metric $I^{(m)} = \mathbb{E}_{x \in \Omega} \mathbb{E}_{i,j}[I_{ij}^{(m)}(x)]$, and $\Delta I^{(m)} \overset{\text{def}}{=} I_{\text{nor}}^{(m)} - I_{\text{adv}}^{(m)}$, which measures the difference in interactions between normal samples and adversarial examples. In this section of the supplementary material, we prove the following property of the metric $\Delta I^{(m)}$.

$$\Delta I^{(m)} = \mathbb{E}_{x \in \Omega} \mathbb{E}_{i,j}[\Delta I_{ij}^{(m)}(x)] \tag{8}$$

where $\Delta I_{ij}^{(m)}(x) = I_{ij}^{(m)}(x) - I_{ij}^{(m)}(x^{\text{adv}})$.

- *Proof*:

$$
\begin{aligned}
\Delta I^{(m)} &= I_{\text{nor}}^{(m)} - I_{\text{adv}}^{(m)} \\
&= \mathbb{E}_{x\in\Omega_{\text{nor}}}\mathbb{E}_{i,j}[I_{ij}^{(m)}(x)] - \mathbb{E}_{x\in\Omega_{\text{adv}}}\mathbb{E}_{i,j}[I_{ij}^{(m)}(x)] \\
&= \mathbb{E}_{x\in\Omega_{\text{nor}}}\mathbb{E}_{i,j}[I_{ij}^{(m)}(x)] - \mathbb{E}_{x\in\Omega_{\text{nor}}}\mathbb{E}_{i,j}[I_{ij}^{(m)}(x+\Delta x)] \\
&= \mathbb{E}_{x\in\Omega_{\text{nor}}}\mathbb{E}_{i,j}[I_{ij}^{(m)}(x) - I_{ij}^{(m)}(x+\Delta x)] \\
&= \mathbb{E}_{x\in\Omega_{\text{nor}}}\mathbb{E}_{i,j}[I_{ij}^{(m)}(x) - I_{ij}^{(m)}(x^{\text{adv}})] \\
&= \mathbb{E}_{x\in\Omega_{\text{nor}}}\mathbb{E}_{i,j}[\Delta I_{ij}^{(m)}(x)]
\end{aligned}
$$

$\square$

# E   Relationship between $\hat{v}(S)$ and $v(S)$

In Section 4.1 of the main paper, we claim that the trend of $v(S)$ can roughly reflect the negative trend of $\hat{v}(S)$. In this section, we discuss the negative correlation between $\hat{v}(S)$ and $v(S)$.

According to Proposition 1, $\hat{v}(S) = H(Y|X_S)$ denotes the entropy of the classification probability given variables in $S$ of the image $x$. Thus, $\hat{v}(S)$ measures the uncertainty of the prediction. If the model prediction is correct and confident, *i.e.* the value of $v(S) = \log p(y = y^{\text{truth}}|x, S)$ is large, then the uncertainty $\hat{v}(S)$ is very low. In comparison, if the model prediction is correct but with a small value of $v(S)$, then the uncertainty is large, yielding a large value of $\hat{v}(S)$. Therefore, the trend of $v(S)$ can roughly reflect the negative trend of $\hat{v}(S)$ when the model prediction is correct.

# F   More discussions about the disentanglement metric

This section provides more discussions about the disentanglement metric in Eq. (4) of the main paper. The motivation of the disentanglement is to measure the discrimination power of interactions of a specific order, as discussed in Section 4.2 of the paper. According to the efficiency property of the multi-order interactions, the model output can be decomposed into the weighted sum of massive interaction components $\Delta v(i, j, S|x)$:

$$
v(N|x) = v(\emptyset|x) + \sum_{i\in N}\phi^{(0)}(i|x) + \sum_{i,j\in N, i\neq j}\sum_{m=0}^{n-2}\frac{n-1-m}{n(n-1)}\mathbb{E}_{S\subseteq N\setminus\{i,j\},|S|=m}[\Delta v(i,j,S)] \quad (9)
$$

In this way, if interaction components of a certain order $m$ are all positive (or negative) and do not conflict with each other, it indicates that these components jointly promote or suppress the model output, showing a strong discrimination power. Otherwise, if some interaction components are positive and others are negative, their effects on the model output will be eliminated. In this case, the discrimination power of interactions is poor.

In this way, we design the disentanglement metric $D^{(m)}$ in Eq. (4) to model the above phenomenon. The physical meaning of this metric is shown in the following equation.

$$
D^{(m)} = \mathbb{E}_{x\in\Omega}\mathbb{E}_{i,j\in N, i\neq j}\frac{\overbrace{|\mathbb{E}_{S\subseteq N\setminus\{i,j\},|S|=m}\Delta v(i,j,S|x)|}^{\text{the strength of the average utility of all components}}}{\mathbb{E}_{S\subseteq N\setminus\{i,j\},|S|=m}\underbrace{|\Delta v(i,j,S|x)|}_{\text{the strength of each component}}} \quad (10)
$$

The numerator measures the strength of the average utility of all interaction components between $(i, j)$ under different contexts $S$. The dominator represents the average strength of each interaction component. If $D^{(m)}$ approximates to 1, then it indicates that almost all interaction components have similar effects (either positive or negative) on the model output. If $D^{(m)}$ approximates to 0, then it shows that most interaction components conflict with each other and are eliminated. Therefore, the disentanglement $D^{(m)}$ measures the discrimination power of interactions.

## G  The attribution-based method of detecting adversarial examples

In Section 4.3 of the paper, we claim that the attribution score used in [23] to detect adversarial exmaples can be writtern as $\phi^{(n-1)}(i|x)$. In the supplementary material, this section provides proofs for this claim.

Yang et al. [23] proposed an attribution-based method to detect adversarial examples, which used the attribution score of input variables. The attribution score of the variable $i$ in [23] is defined as

$$\phi(x)_i := f(x)_c - f(x_{(i)})_c, \quad \text{where} \quad c = \arg\max_{j \in C} f(x)_j \tag{11}$$

where $x$ denotes the original input sample, and $x_{(i)}$ denotes the input sample with the $i$-th variable masked by 0. $f(x)_c$ denotes the network output of the $c$-th category. Actually, $f(x)_c$ can also be written as $v(N|x)$, and $f(x_{(i)})_c$ can be written as $v(N \setminus \{i\}|x)$. Thus, the attribution score can be represented as $v(N|x) - v(N \setminus \{i\}|x)$. We prove that $v(N|x) - v(N \setminus \{i\}|x) = \phi^{(n-1)}(i|x)$.

• *Proof*:

$$\begin{aligned}
v(N|x) - v(N \setminus \{i\}|x) &= v((N \setminus \{i\}) \cup \{i\}|x) - v(N \setminus \{i\}|x) \\
&= v(S \cup \{i\}|x) - v(S|x) \qquad \% \; S \stackrel{\text{def}}{=} N \setminus \{i\}, |S| = n - 1 \\
&= \mathbb{E}_{S \subseteq N \setminus \{i\}, |S|=n-1} \left[ v(S \cup \{i\}|x) - v(S|x) \right] \\
&= \phi^{(n-1)}(i|x)
\end{aligned}$$

$\square$

According to the *accumulation property* of multi-order Shapley values and interactions, we have $\phi^{(n-1)}(i|x) = \mathbb{E}_{j \in N \setminus \{i\}} \left[ \sum_{m=0}^{n-2} I_{ij}^{(m)} \right] + \phi^{(0)}(i|x)$. This indicates that $\phi^{(n-1)}(i|x)$ contains the interaction components with the highest order ($m = n - 2$), which are not included in Shapley values with orders lower than $n - 1$. Section 4.1 of the paper has pointed that high-order interactions are the most sensitive to adversarial perturbations, thereby enabling the detection of adversarial examples.

## H  Effectiveness of the dropout method to alleviate adversarial utilities

In Section 4.3 of the paper, we claim that the dropout operation mainly hurts high-order interactions, rather than low-order interactions. In this section, we theoretically prove such effects of the dropout operation.

Given the input sample $x \in \mathbb{R}^n$ and the dropout rate $\alpha$, let $\mathcal{K} = \{K | K \subset N, |K| = \lfloor (1 - \alpha)n \rfloor\}$ denote all possible sets of remained variables after the dropout operation. Let $v^\alpha(N|x) = \mathbb{E}_{K \in \mathcal{K}}[v(K|x)]$ denote the average network output among all inputs after the dropout operation with rate $\alpha$. According to the efficiency property of the multi-order interaction, we have

$$v(N|x) = v(\emptyset|x) + \sum_{i \in N} \phi^{(0)}(i|x) + \sum_{i \neq j \in N} \sum_{m=0}^{n-2} \frac{n-1-m}{n(n-1)} I_{ij,N}^{(m)}(x) \tag{12}$$

where $I_{ij,N}^{(m)}(x)$ denotes the $m$-order interaction between variables $(i, j)$ of the input $x$ with all variables $N$. Similarly,

$$v(K|x) = v(\emptyset|x) + \sum_{i \in K} \phi^{(0)}(i|K, x) + \sum_{i \neq j \in K} \sum_{m=0}^{k-2} \frac{k-1-m}{k(k-1)} I_{ij,K}^{(m)}(x) \tag{13}$$

where $k = |K| = \lfloor (1-\alpha)n \rfloor$. Thus,

$$v^\alpha(N|x) = \mathbb{E}_{K \in \mathcal{K}}[v(K|x)]$$

$$= \mathbb{E}_{K \in \mathcal{K}}\left[ v(\emptyset|x) + \sum_{i \in K} \phi^{(0)}(i|K,x) + \sum_{i \neq j \in K} \sum_{m=0}^{k-2} \frac{k-1-m}{k(k-1)} I_{ij,K}^{(m)}(x) \right]$$

$$= v(\emptyset|x) + \mathbb{E}_{K \in \mathcal{K}}\left[ \sum_{i \in K}(v(i|x) - v(\emptyset|x)) \right] + \mathbb{E}_{K \in \mathcal{K}}\left[ \sum_{i \neq j \in K} \sum_{m=0}^{k-2} \frac{k-1-m}{k(k-1)} I_{ij,K}^{(m)}(x) \right]$$

$$= v(\emptyset|x) + \mathbb{E}_{\substack{K \subset N \\ k=(1-\alpha)n}}\left[ \sum_{i \in K} \phi^{(0)}(i|x) \right] + \mathbb{E}_{\substack{K \subset N \\ k=(1-\alpha)n}}\left[ \sum_{i \neq j \in K} \sum_{m=0}^{k-2} \frac{k-1-m}{k(k-1)} I_{ij,K}^{(m)}(x) \right]$$

$$= v(\emptyset|x) + (1-\alpha) \sum_{i \in N} \phi^{(0)}(i|x) + \mathbb{E}_{\substack{K \subset N \\ k=(1-\alpha)n}}\left[ \sum_{i \neq j \in K} \sum_{m=0}^{k-2} \frac{k-1-m}{k(k-1)} I_{ij,K}^{(m)}(x) \right]$$

$$= v(\emptyset|x) + (1-\alpha) \sum_{i \in N} \phi^{(0)}(i|x) + \mathbb{E}_{\substack{K \subset N \\ k=(1-\alpha)n}}\left[ \sum_{i \neq j \in K} \sum_{m=0}^{k-2} \frac{k-1-m}{k(k-1)} \mathbb{E}_{\substack{S \subseteq K \setminus \{i,j\} \\ |S|=m}} [\Delta v(i,j,S|x)] \right]$$

$$= v(\emptyset|x) + (1-\alpha) \sum_{i \in N} \phi^{(0)}(i|x) + \frac{k(k-1)}{n(n-1)} \sum_{i \neq j \in N} \sum_{m=0}^{k-2} \frac{k-1-m}{k(k-1)} \mathbb{E}_{\substack{K \subset N \\ k=(1-\alpha)n}} \mathbb{E}_{\substack{S \subseteq K \setminus \{i,j\} \\ |S|=m}} [\Delta v(i,j,S|x)]$$

$$= v(\emptyset|x) + (1-\alpha) \sum_{i \in N} \phi^{(0)}(i|x) + \sum_{i \neq j \in N} \sum_{m=0}^{k-2} \frac{k-1-m}{n(n-1)} \mathbb{E}_{\substack{K \subset N \\ k=(1-\alpha)n}} \mathbb{E}_{\substack{S \subseteq K \setminus \{i,j\} \\ |S|=m}} [\Delta v(i,j,S|x)]$$

$$= v(\emptyset|x) + (1-\alpha) \sum_{i \in N} \phi^{(0)}(i|x) + \sum_{i \neq j \in N} \sum_{m=0}^{k-2} \frac{k-1-m}{n(n-1)} \mathbb{E}_{\substack{S \subset N \setminus \{i,j\} \\ |S|=m}} [\Delta v(i,j,S|x)]$$

$$= v(\emptyset|x) + (1-\alpha) \sum_{i \in N} \phi^{(0)}(i|x) + \sum_{i \neq j \in N} \sum_{m=0}^{k-2} \frac{k-1-m}{n(n-1)} I_{ij,N}^{(m)}(x)$$

$$\tag{14}$$

Thus, the change in the network output caused by the dropout operation can be represented as follows,

$$v(N|x) - v^\alpha(N|x) = \alpha \sum_{i \in N} \phi^{(0)}(i,x) + \sum_{i \neq j \in N}\left[ \sum_{m=0}^{n-2} \frac{n-1-m}{n(n-1)} I_{ij,N}^{(m)}(x) - \sum_{m=0}^{k-2} \frac{k-1-m}{n(n-1)} I_{ij,N}^{(m)}(x) \right]$$

$$= \alpha \sum_{i \in N} \phi^{(0)}(i,x) + \sum_{i \neq j \in N} \sum_{m=0}^{k-2} \frac{n-k}{n(n-1)} I_{ij,N}^{(m)}(x) + \sum_{i \neq j \in N} \sum_{m=k-1}^{n-2} \frac{n-1-m}{n(n-1)} I_{ij,N}^{(m)}(x)$$

$$= \alpha \sum_{i \in N} \phi^{(0)}(i,x) + \frac{\alpha}{n-1} \sum_{i \neq j \in N} \sum_{m=0}^{k-2} I_{ij,N}^{(m)}(x) + \underbrace{\sum_{i \neq j \in N} \sum_{m=k-1}^{n-2} \frac{n-1-m}{n(n-1)} I_{ij,N}^{(m)}(x)}_{\text{high-order interactions}}$$

$$\tag{15}$$

According to Eq. (15), the dropout operation removes all high-order interactions $(m > (1-\alpha)n - 2)$, while slightly affects low-order interactions. Thus, the dropout operation can remove sensitive interaction components of the DNN, thereby reducing the attacking utility of perturbations and correcting the network output.

## I  More experimental results

### I.1  More visualization results based on ResNet-50

The method in [25] could be directly extended to visualize salient concepts of the multi-order interaction without much change. For a specific order $m$, we visualized all contexts $\{S\}$ of $I_{ij}^{(m)}$ for each pair $(i,j)$ with top $10\%$ interaction strengths. We exclusively visualized contexts $\{S\}$ that boosted the strength of $I_{ij}^{(m)}$, i.e. $\Delta v(S,i,j) \cdot I_{ij}^{(m)} > 0$. Let map$(S) \in \{0,1\}^n$ denote the

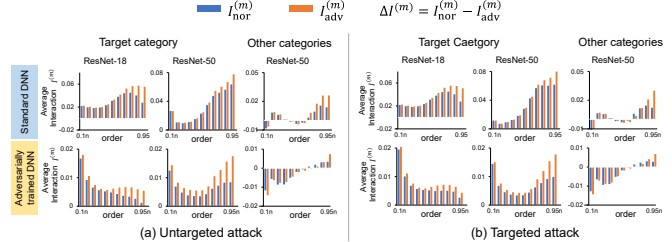

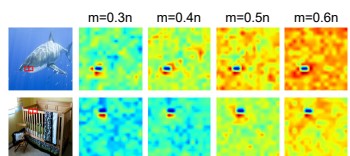

Figure 1: Interactions *w.r.t.* the target category and other categories.

Figure 2: Contexts of the m-order interaction $I_{ij}^{(m)}$ in normal samples of standard ResNet-50.

Table 1: Interaction behaviors in standard DNNs and adversarially trained DNNs are quite different *w.r.t.* the metric $\frac{|\Delta I^{(m)}|}{\sum_{m'} |\Delta I^{(m')}|}$.

| $m$ | $0.1n$ | $0.2n$ | $0.3n$ | $0.4n$ | $0.5n$ | $0.6n$ | $0.7n$ | $0.8n$ | $0.85n$ | $0.9n$ | $0.95n$ |
|---|---|---|---|---|---|---|---|---|---|---|---|
| $\frac{|\Delta I^{(m)}|}{\sum_{m'} |\Delta I^{(m')}|}$ in standard ResNet-18 | 0.003 | 0.005 | 0.008 | 0.014 | 0.021 | 0.040 | 0.066 | 0.103 | **0.138** | **0.219** | **0.383** |
| $\frac{|\Delta I^{(m)}|}{\sum_{m'} |\Delta I^{(m')}|}$ in adversarially trained ResNet-18 | **0.016** | **0.023** | **0.055** | **0.082** | **0.095** | **0.100** | **0.100** | **0.108** | 0.117 | 0.144 | 0.160 |

map corresponding to the context $S$. If the $k$-th pixel was contained in $S$, then $\text{map}_k(S) = 1$; otherwise, $\text{map}_k(S) = 0$. In this way, we visualized the weighted average contexts *w.r.t.* pixels $(i, j)$ as $\sum_{S \subseteq N, |S|=m} |\Delta v(S, i, j)| \cdot \text{map}(S)$.

Figure 2 (right) in the paper has shown visualization results on ResNet-18. Here, we presented more visualization results of interaction contexts based on ResNet-50 [8] in Figure 2. As Figure 2 shows, low-order interactions usually represented simple features of local collaborations, and high-order interactions usually reflected complex features of global collaborations.

## I.2 More discussions about the difference between standard DNNs and adversarially trained DNNs in Figure 3 of the main paper

In Figure 3 of the main paper, it seems that standard DNNs and adversarially trained DNNs were similar under the metric $\Delta I^{(m)}$, but Figure 3 does reflect the significant difference of interaction behaviors between standard DNNs and adversarially trained DNNs. Figure 3 shows that although high-order interactions were penalized in both standard and adversarially trained DNNs, high-order interactions in the standard DNN decreased much more than those in the adversarially trained DNN.

More specifically, we define the metric $\frac{|\Delta I^{(m)}|}{\sum_{m'} |\Delta I^{(m')}|}$ to compare the change of multi-order interactions in different DNNs, where $\Delta I^{(m)} = I_{\text{nor}}^{(m)} - I_{\text{adv}}^{(m)}$ measures the difference of the $m$-th order interaction between normal samples and adversarial examples. $\sum_{m'} |\Delta I^{(m')}|$ is the total strength of differences of multi-order interactions, which is used for normalization. Table 1 shows that the interaction behaviors in standard DNNs and adversarially trained DNNs were quite different. In the standard ResNet-18, low-order interactions were penalized slightly, and high-order interactions dropped a lot. In comparison, in the adversarially trained ResNet-18, low-order interactions and middle-order interactions were penalized more than those in the standard ResNet-18, while high-order interactions were less penalized than those in the standard ResNet-18. Therefore, the adversarially trained DNN was supposed to exhibit a larger value of $\frac{|\Delta I^{(m)}|}{\sum_{m'} |\Delta I^{(m')}|}$ for low-order interactions (with small $m$) than the standard DNN. In comparison, the adversarially trained DNN exhibited a smaller value of $\frac{|\Delta I^{(m)}|}{\sum_{m'} |\Delta I^{(m')}|}$ for high-order interactions (with large $m$) than the standard DNN.

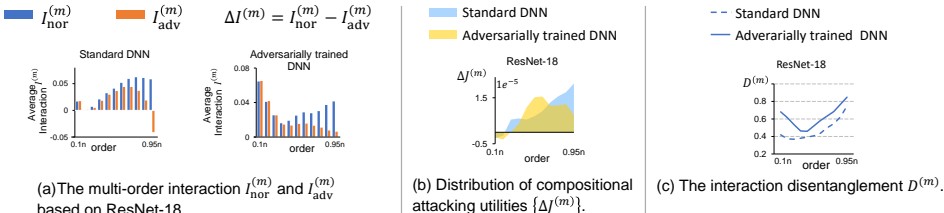

Figure 3: Experimental results on ResNet-18 when setting the masked variables to zero.

## I.3 Discussions about attacking utilities of multi-order interactions.

Figure 4 of the paper shows that low-order interactions in adversarially trained DNNs usually have more attacking utilities. Note that the attacking utility on high/middle-order interactions may make $\Delta J^{(m)}$ on low-order interactions negative, as a trade-off. According to Eq. (3), the sum of attacking utilities on low-order, middle-order, and high-order interactions is a constant ($\Delta v(N|x)$). Therefore, the penalization of high-order interactions may also cause the increase of low-order interactions to some extent. In this way, there are two effects on low-order interactions. First, adversarial attacks penalize low-order interactions. Second, the penalization of high-order interactions also boosts low-order interactions as a side effect. In such a trade-off, low-order interactions decrease in most DNNs. In this way, low-order interactions are increased in a few special cases, which can be explained by the above analysis.

## I.4 More details about the approximation method of computing interactions

To reduce the computational cost, we applied the sampling-based approximation method in [25] and did not compute interactions at the pixel-wise level. Instead, we split the image into $16 \times 16$ grids, and took each grid as a single input variable, thereby $n = 256$. Then, we randomly sampled 200 pairs of grids $(i, j)$, and for each pair of grids and each order $m$, we sampled contexts $S$ s.t. $|S| = m$ for 100 times to approximate the interaction. Then we computed the $m$-th order interactions based on Eq. (2) in the main paper by using the sampled pairs of grids and contexts.

## I.5 More details about the setting of masking in the computation of interactions

In the main paper, we measured the interaction $I^{(m)}$ by setting $v(S|x) = \log p(y = y^{\text{truth}}|$ given variables in $S$ in the input $x$ and mask variables in $N \setminus S$). Variables not in $S$ were set to the average value over different input samples following settings in [1], to represent their absence. In this section, we conducted experiments by using setting masked variables to zero, to verify that the effects of the choice of masking on the results does not affect our conclusions. Figure 3 shows that the choice of masking did not affect the conclusion that adversarial attacks mainly affect high-order interactions.

## I.6 Analysis about interactions *w.r.t.* the target category and other categories

In the main paper, experimental results on the ground-truth category have verified our conclusions that adversarial attacks mainly affect high-order interactions. In this section of the supplementary material, we show more results on the target category and other categories, which also verified our conclusions.

In the targeted attack, the target category was referred to as the target label $y^{\text{target}} \neq y^{\text{truth}}$. In the untargeted attack, we considered the misclassified category $y^{\text{pred}} \neq y^{\text{truth}}$ as the target category. Given each input image, we measured the interaction *w.r.t.* the output of its target category by setting $v(S|x) = \log p(y = y^{\text{target}}|S, x)$. Besides, we also measured the interactions *w.r.t.* outputs of other categories, except the ground-truth category. Considering the softmax operation $p(y = y^{\text{truth}}|S, x) = \frac{\exp(h_{y^{\text{truth}}}(S|x))}{\sum_{y'} \exp(h_{y'}(S|x))}$, we set $v(S|x) = \log \sum_{y' \neq y^{\text{truth}}} \exp(h_{y'}(S|x))$ to measure the interaction *w.r.t.* effects on other categories. $h_{y'}(S|x)$ denoted the network output of the category $y'$ before the softmax layer, when we took variables in $S$ of $x$ as the input. Figure 1 shows interactions *w.r.t.* outputs of the target category and other categories. We found that high-order interactions *w.r.t.* other categories usually increased. Meanwhile, high-order interactions *w.r.t.* the target category also

Table 2: The close relationship between the attacking strength and changes of high-order interactions.

| Attacking strength (iteration number) | $\Delta I^{(0.7n)}$ | $\Delta I^{(0.8n)}$ | $\Delta I^{(0.85n)}$ | $\Delta I^{(0.9n)}$ | $\Delta I^{(0.95n)}$ |
|---|---|---|---|---|---|
| 4 | 0.0518 | 0.0990 | 0.1411 | 0.1944 | **0.2274** |
| 8 | 0.0789 | 0.1741 | 0.2399 | 0.3118 | **0.3632** |
| 16 | 0.1113 | 0.2565 | 0.3342 | 0.4157 | **0.4657** |
| 32 | 0.1512 | 0.3338 | 0.4391 | 0.5480 | **0.6085** |
| Pearson correlation coefficient | 0.9799 | 0.9603 | 0.9635 | 0.9667 | 0.9634 |

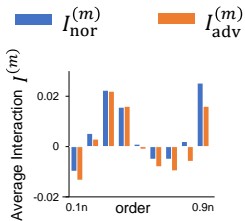

(a) The multi-order interaction $I_{nor}^{(m)}$ and $I_{adv}^{(m)}$ based on the ResNet-50 trained via randomized smoothing.

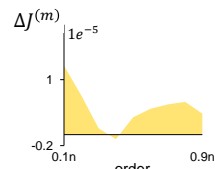

(b) Distribution of compositional attacking utilities $\{\Delta J^{(m)}\}$ on the ResNet-50 trained via randomized smoothing.

Figure 4: Experimental results on the ResNet-50 trained via randomized smoothing.

significantly increased. This indicated that adversarial perturbations adversely affected complex features corresponding to the ground-truth category, while encouraging features for other categories, especially for the target category.

## I.7 Extended experiments on the relationship between the strength of attacks and the change of high-order interactions

In Section 4.1 of the paper, we find that adversarial attacks mainly affected high-order interactions in DNNs. In this section, we further explore the relationship between the strength of attacks and the change of high-order interactions.

Using the standard ResNet-18 learned on the ImageNet dataset, we conducted untargeted PGD attacks on input samples with different attacking strengths, in order to test the effects of attacking strength on high-order interactions. The strength of attacks was represented by the iteration numbers of the PGD attack. Then, we computed the change in high-order interactions $\Delta I^{(m)}$ caused by adversarial attacks of different strengths. We also computed the Pearson correlation coefficient between the strength of attacks and $\Delta I^{(m)}$. Table 2 shows a close relationship between changes in high-order interactions and the strength of adversarial attacks. Besides, for both weak and strong attacks, the highest-order (here we set it as $0.95n$-order) interactions were much more sensitive than interactions of not-so-high orders (e.g., $0.7n$-order).

## I.8 Extended experiments based on models with certified robustness

In the main paper, experimental results on DNNs trained via adversarial training [12] have verified our conclusions that adversarial attacks mainly affect high-order interactions, and low-order interactions in adversarial examples towards robust DNNs have more attacking utilities. In this section, we conducted extended experiments on other defending method [2]. Given the ResNet-50 trained via randomized smoothing, which was proved to have certified robustness by [2], we generated adversarial examples using the PGD attack on samples from the validation set of the ImageNet dataset. We followed experimental settings of the classic PGD attack (see Section 4.1 in the main paper). Then, we computed multi-order interactions in normal samples and adversarial examples on the pre-trained ResNet-50.

Figure 4 shows that in the model trained via randomized smoothing, adversarial attacks also mainly affected high-order interactions, and low-order interactions had more attacking utilities on the model output, which verified our conclusions.

Table 3: The distance between normal samples $x$ and adversarial examples $x^{\text{adv}}$, and the distance between $x$ and the recovered samples $\hat{x}$.

| | Standard DNN | | Adversarially trained DNN | |
|---|---|---|---|---|
| | $\mathbb{E}\|x - x^{\text{adv}}\|_2$ | $\mathbb{E}\|x - \hat{x}\|_2$ | $\mathbb{E}\|x - x^{\text{adv}}\|_2$ | $\mathbb{E}\|x - \hat{x}\|_2$ |
| ResNet-18 | **9.72** | 13.57 | 18.69 | **11.45** |
| ResNet-50 | **9.56** | 13.40 | 18.34 | **12.68** |
| DenseNet-161 | **9.67** | 13.51 | 18.55 | **13.26** |

## I.9 More discussions about the comparison with frequency-based methods

In Section 4.1 of the main paper, we have compared our interaction-based metric with the frequency-based metric. This section provides more discussions about the comparative experiment.

Previous studies [24, 20, 7] explained adversarial perturbations as high-frequency features. In this paper, we compared the frequency metric with the interaction metric, and showed that our interaction metric could better explain the essential property of adversarial perturbations. In order to measure the significance of features of different frequencies in an input image $x$, we applied the Fourier transform to the image to obtain the frequency spectrum $F$. Then, we shifted the low frequency components to the center of the spectrum. The magnitude of the $r$-frequency component was quantified as $F^{(r)} = \sqrt{\sum_{(h,w):h^2+w^2=r^2} \|F^{(h,w)}\|_2^2}$. In order to fairly compare these two metrics, we computed the Fisher's discriminant ratio [5] using two metrics, which measured the ratio of the variance between the classes (between normal samples and adversarial examples) to the variance within the classes. Such experimental settings has also been introduced in Figure 6 (left) of the main paper.

## I.10 Explaining high recoverability of adversarial examples on adversarially trained DNNs

In Section 4.3 of the paper, we claim that adversarial examples towards adversarially trained DNNs usually exhibit higher recoverability than adversarial examples towards standard DNNs. In the supplementary material, this section shows experimental results to verify the above claim.

We generated adversarial examples for normal validation samples in the ImageNet dataset by following settings of the untargeted PGD attack [12], in which $\epsilon = 16/255$, and the attack was conducted with 10 steps with the step size $2/255$. Adversarial examples were generated based on ResNet-18/50 and DenseNet-161 trained on the ImageNet dataset. Then, we used same parameters to conduct the targeted PGD attack and recover normal samples.

Table 3 shows that adversarially trained DNNs usually exhibited higher recoverability than standard DNNs. This can be explained Proposition 1 in the paper. As we have discussed in Section 4.1 of the paper, adversarial perturbations towards adversarially trained DNNs usually pay more attention to low-order interactions than perturbations towards standard DNNs. On the other hand, the low-order interaction $I_{ij}^{(m)}(x)$ is equivalent to the conditional mutual information $MI(X_i; X_j; Y|X_S)$ given small contexts $X_S$ according to Proposition 1, while the high-order interaction corresponds to such a mutual information conditioned on large contexts with massive variables. In general, compared to high-order interactions, low-order interactions are conditioned on less contextual variables, thereby suffering less from adversarial perturbations, *i.e.* obviously $MI(X_i; X_j; Y|X_S)$ usually suffers less from adversarial perturbations, if the condition $x_S$ only contains very few variables. In other words, low-order interactions are more transferable among different contexts $x_S$, so it is easy to invert adversarial perturbations of low-order interactions. In this way, because adversarial perturbations for adversarially trained DNNs mainly focus on low-order interactions, such perturbations are easy to be recovered.