# OpenReview forum: "Towards a Unified Game-Theoretic View of Adversarial Perturbations and Robustness"
_NeurIPS.cc/2021/Conference — NeurIPS 2021 Poster_

### Official Review · Reviewer_Razy · 2021-07-15

**Rating:** 7
**Confidence:** 3

**Summary:**

This work provides a unified understanding for adversarial perturbations and robustness based on the multi-order interaction between input variables.  The experimental results and theoretical analyses validate that adversarial perturbations are highly correlated to the high-order interactions. Adversarial training significantly improves the disentanglement of low-order interactions, thus enhancing the robustness of sensitive high-order interactions.  The proposed multi-order perspective of adversarial robustness can explain the reasons for some methods  improving adversarial robustness.

**Ethical Concerns:**

No ethical issues.

**Limitations And Societal Impact:**

This paper contributes to understanding adversarial robutness. Therefore,  no potential negative societal impacts incur.

**Main Review:**

This paper is well written and organized. The understanding for adversarial robustness seems to be well motivated and brings much conceptual improvement. The research studies the relationship between the adversarial attacks and robustness, and multi-order interactions.
On one hand, the research discovers high-order interactions are frequently affected by adversarial attacks and explains this phenomenon theoretically. On the other hand, the authors show that adversarially-trained DNNs make the low-order interactions more discriminative compared to naturally-trained DNNs. Comprehensive experiments are conducted and support the claims in this paper. Besides, this paper provides the unified understanding for the success of some existing adversarial training method from this novel perspective.

There is a minor question --- I wonder if the strength of adversarial attacks (e.g., the iteration numbers of PGD) is related to the perturbations on high-order interations. That is to say, could we use the pertubetions on high-order interactions to measure the strength of adversarial attacks?

**Time Spent Reviewing:**

2

---

> ### Author Response · Authors · 2021-08-10
> **Response to Reviewer (Razy)**
>
> Thanks for your positive feedback and valuable comments on the paper.
>
> Q1: " I wonder if the strength of adversarial attacks (e.g., the iteration numbers of PGD) is related to the perturbations on high-order interactions. That is to say, could we use the perturbations on high-order interactions to measure the strength of adversarial attacks?"
>
> A1: Thank you. We have followed your suggestions to conduct **a new experiment** to analyze the relationship between changes in high-order interactions and the strength of adversarial attacks. Experimental results show that changes in high-order interactions can indicate the strength of adversarial attacks. Using the standard ResNet-18 learned on the ImageNet dataset, we conducted untargeted PGD attacks on input samples with different attacking strengths, in order to test the effects of attacking strength on high-order interactions. The strength of attacks was represented by the iteration numbers of the PGD attack. Then, we computed the change in high-order interactions $\Delta I^{(m)}$​​​​​​​ caused by adversarial attacks of different strengths. We also computed the Pearson correlation coefficient between the strength of attacks and $\Delta I^{(m)}$​​​​​​​​​​​​​. The following results show a close relationship between changes in high-order interactions and the strength of adversarial attacks.
>
> | Attacking strength (iteration number) | 4      | 8      | 16     | 32         | Pearson correlation coefficient |
> | ------------------------------------- | :----- | :----- | :----- | :--------- | :-----------------------------: |
> | $\Delta I^{(0.7n)}$                   | 0.0518 | 0.0789 | 0.1113 | **0.1512** |             0.9799              |
> | $\Delta I^{(0.8n)}$                   | 0.0990 | 0.1741 | 0.2565 | **0.3338** |             0.9602              |
> | $\Delta I^{(0.85n)}$                  | 0.1411 | 0.2399 | 0.3342 | **0.4391** |             0.9635              |
> | $\Delta I^{(0.9n)}$                   | 0.1944 | 0.3118 | 0.4175 | **0.5480** |             0.9667              |
> | $\Delta I^{(0.95n)}$                  | 0.2274 | 0.3632 | 0.4657 | **0.6085** |             0.9634              |
>
> By the way, for both weak and strong attacks, the highest-order (here we set it as $0.95n$​-order) interactions are much more sensitive than interactions of not-so-high orders (*e.g.*, $0.7n$​-order).
>
> ---

---

> > ### Comment · Reviewer_Razy · 2021-08-31
> > **Thank you for your response**
> >
> > Thank you for the detailed answers. The answer to Q1 is satisfactory. Therefore, I am willing to keep my score after rebuttal.

---

### Official Review · Reviewer_qpWF · 2021-07-16

**Rating:** 6
**Confidence:** 3

**Summary:**

The authors discover and partially prove that adversarial perturbations mainly affect high-order interactions, rather than low-order interactions.

The authors find that there is a clear difference between standard DNNs and adversarially trained DNNs. Adversarial training significantly increases the robustness of high-order interactions.

The research provides a unified understanding for the success of several existing adversarial defense methods,

**Limitations And Societal Impact:**

The authors do not mention this.

**Main Review:**

There is one related paper which also performs a game theoretic analysis for adversarial attacks and defenses [*1]. The authors should make a comparison with the paper.

The authors define the disentanglement metric to identify whether interactions of this order are discriminative for the classification of a specific category, or represent common knowledge shared by different categories. Based on the disentanglement metric, the authors discover that compared with standard DNNs, adversarially trained DNNs usually encode more discriminative low-order interactions. Discriminative low-order interactions make high order interactions of adversarially trained DNNs robust to attacks.

Some studies explained adversarial examples as high-frequency features and high-rank features. It is glad to see that the problem can be viewed from a different aspect. The high order interactions can better explain the essential property of attacking-sensitive representations, i.e. the complex and large-scale visual concepts.

[*1] A Game Theoretic Analysis of Additive Adversarial Attacks and Defenses, NeurIPS 2020

**Time Spent Reviewing:**

6

---

> ### Author Response · Authors · 2021-08-10
> **Response to Reviewer (qpWF)**
>
> Thanks for your positive feedback and valuable comments on the paper.
>
> Q1: "There is one related paper which also performs a game theoretic analysis for adversarial attacks and defenses [cite 1]. The authors should make a comparison with the paper."
>
> A1: Thank you for recommending such an interesting paper to us. We will follow your suggestions to cite [cite 1] and discuss the advantages and essential differences between [cite 1] and our work. [cite 1] proposed a game-theoretic framework to formulate the game between attackers and defenders of a given DNN. Under the game-theoretic framework, authors theoretically proved that the FGM attack and the random smoothing defense formed a Nash Equilibrium under some assumptions. We think this paper provided new insights into understanding the adversarial robustness of DNNs.
>
> Unlike [cite 1], we formulate the inference process of a DNN as a game, and analyze the relationship between input variables  (*e.g.*, pixels in images) of the DNN from the game-theoretic perspective. We focus on the difference of game-theoretic interactions between normal samples and adversarial examples. From this perspective, we find that adversarial attacks mainly affect high-order interactions, and adversarially training boosts the robustness of high-order interactions. Furthermore, as discussed in Section 2, many adversarial defending methods are defined in a heuristic manner without theoretical supports. To this end, our research summarizes and refines the essential and common mechanism of such defending methods from the perspective of game-theoretic interactions.
>
> [cite 1] Pal, Ambar, and Rene Vidal. A Game Theoretic Analysis of Additive Adversarial Attacks and Defenses. In Advances in Neural Information Processing Systems, 2020.

---

> > ### Comment · Reviewer_qpWF · 2021-08-22
> > **Keep score after rebuttal**
> >
> > I went over the rebuttal. I have no more questions and would keep the score after rebuttal.

---

### Official Review · Reviewer_T63V · 2021-07-16

**Rating:** 4
**Confidence:** 3

**Summary:**

The Shapley interaction index is leveraged to analyze the adversarial robustness, with an ambitious goal of unifying different explanations for adversarial attacks and defenses.

**Ethical Concerns:**

N.A.

**Limitations And Societal Impact:**

The authors may improve the writing by narrowing down the scope.

**Main Review:**

Originality:

a. The Shapley (interaction) index is a well-known mathematical quantity in the field of game theory and has been explored in the ML community, particularly to understand DNN [1].

b. A similar decomposition of the Sharpley interaction index has also been provided in [58], which may undermine the technical novelty of the current work.

c. The discovery that higher-order interactions are affected by the adv attack the most is interesting.

Quality:

a. From Figure 3, it seems to the reviewer that the behaviors of standard and adv trained CNNs are quite similar under the $I^{(m)}$ measure. Does this mean that the adv training is not that effective? If so, this goes against empirical evidence that adv training is helpful?

b. Have the authors tried to experiment with the training methods with certified robustness? The reviewer wonders whether a similar figure (like figure 3) can be produced.


Clarity:

a. Overall, the reviewer is comfortable reading the first two sections and begins to find the remaining sections difficult to comprehend. For example, why can the trend of $\hat{v}(S)$ be reflected by $v(S)$?

b. What are the motivations of the two equations in Eq. (4)?

c. How do the authors implement the masking? Setting to zero or mid-gray or Gaussian noise? Will different choices affect the results?







**Time Spent Reviewing:**

5

---

> ### Author Response · Authors · 2021-08-10
> **Response to Reviewer (T63V) (Part 1)**
>
> Thank you very much for your careful review and constructive comments. We will try our best to answer all your concerns.
>
> Q1: "The Shapley interaction index is a well-known mathematical quantity in the field of game theory and has been explored in the ML community, particularly to understand DNN. A similar decomposition of the Shapley interaction index has also been provided in [58], which may undermine the technical novelty of the current work. The discovery that higher-order interactions are affected by the adv attack the most is interesting ."
>
> A1: In this paper, our theoretical novelty is that we theoretically prove that the decomposition of Shapley interaction index (the multi-order interaction) is an appropriate metric to analyze the adversarial robustness of DNNs. Although Zhang et al. [58] have proposed the similar decomposition of the Shapley interaction index, it is still questionable to directly use this metric to fairly analyze the robustness of DNNs. This is because there is no theory to prove the correctness and fairness of using this metric to analyze adversarial perturbations and defense. To this end, we take the first step to prove some important properties of this interaction metric, which enable this metric to be used to fairly analyze the adversarial robustness of DNNs.
>
> (1) We have proven that the network output can be decomposed as the weighted sum of multi-order interactions. Please see Lines 146-149 and Lines 165-167.
>
> (2) We have also proven *linearity, nullity, commutativity,* and *symmetry* properties of the multi-order interaction. Please see Lines 162-164 and the supplementary material for details.
>
> (3) Furthermore, we have proven the equivalence between the multi-order interaction and mutual information. Please see Proposition 1 in Lines 231-238.
>
> The above properties of the multi-order interaction enable us to analyze the robustness of DNNs from the game-theoretic perspective.
>
> Besides, this paper also has two distinctive contributions. (1) We summarize and refine the essential and common mechanism of some defending methods from the perspective of game-theoretic interactions. (2) We explain the effectiveness of adversarial training from the perspective of game-theoretic interaction. We discover that adversarially training boosts the discrimination power of low-order interactions and the robustness of high-order interactions.
>
> - - -
>
> Q2: "From Figure 3, it seems to the reviewer that the behaviors of standard and adv trained CNNs are quit similar under the measure. Does this mean that the adv training is not that effective?"
>
> A2: Thank you very much for your careful review, but Figure 3 does reflect the significant difference of interaction behaviors between standard DNNs and adversarially trained DNNs. Figure 3 shows that although high-order interactions are penalized in both standard and adversarially trained DNNs, high-order interactions in the standard DNN decreased much more than those in the adversarially trained DNN.
>
> More specifically, we define the metric  $\frac{|\Delta I^{(m)}|}{\sum_{m^\prime} |\Delta I^{(m^\prime)}|}$​​​ to compare the change of multi-order interactions in different DNNs, where $\Delta I^{(m)}=I_{\text{ori}}^{(m)}-I_{\text{adv}}^{(m)}$​​​ measures the difference of the $m$​​​-order interaction between normal samples and adversarial examples. $\sum_{m^\prime} |\Delta I^{(m^\prime)}|$​​​ is the total strength of differences of multi-order interactions, which is used for normalization. The following table shows that the interaction behaviors in standard DNNs and adversarially trained DNNs are quite different. In the standard ResNet-18, low-order interactions are penalized slightly, and high-order interactions drop a lot. In comparison, in the adversarially trained ResNet-18, low-order interactions and middle-order interactions are penalized more than those in the standard ResNet-18, while high-order interactions are less penalized than those in the standard ResNet-18. Therefore, the adversarially trained DNN is supposed to exhibit larger $\frac{|\Delta I^{(m)}|}{\sum_{m^\prime} |\Delta I^{(m^\prime)}|}$​​​ for low-order interactions (with small $m$​) than the standard DNN. In comparison, the adversarially trained DNN exhibits smaller $\frac{|\Delta I^{(m)}|}{\sum_{m^\prime} |\Delta I^{(m^\prime)}|}$​​​​ for high-order interactions (with large $m$​) than the standard DNN.
>
> | m                                                            | 0.1n      | 0.2n      | 0.3n      | 0.4n      | 0.5n      | 0.6n         | 0.7n         | 0.8n         | 0.85n      | 0.9n       | 0.95n      |
> | ------------------------------------------------------------ | :-------- | :-------- | :-------- | :-------- | :-------- | :----------- | :----------- | :----------- | :--------- | :--------- | :--------- |
> |                                                              | low-order | low-order | low-order | low-order | low-order | middle-order | middle-order | middle-order | high-order | high-order | high-order |
> | $\frac{\|\Delta I^{(m)}\|}{\sum_{m^\prime} \|\Delta I^{(m^\prime)}\|}$​​​ in standard ResNet-18 | 0.003     | 0.005     | 0.008     | 0.014     | 0.021     | 0.040        | 0.066        | 0.103        | **0.138**  | **0.219**  | **0.383**  |
> | $\frac{\|\Delta I^{(m)}\|}{\sum_{m^\prime} \|\Delta I^{(m^\prime)}\|}$​​​ in adversarially trained ResNet-18 | **0.016** | **0.023** | **0.055** | **0.082** | **0.095** | **0.100**    | **0.100**    | **0.108**    | 0.117      | 0.144      | 0.160      |
>
> Furthermore, the differences between standard DNNs and adversarially trained DNNs are better reflected in Figure 4 and Figure 5 of the paper. Figure 4 shows that in adversarially trained DNNs, although high-order interactions are usually sensitive to attacks, low/middle-order interactions take up more attacking utilities. Figure 5 shows that interactions of adversarially trained DNNs are more disentangled than those of standard DNNs, especially for low-order interactions.  This indicates that low-order interactions in adversarially trained DNNs encode more category-specific information for inference than low-order interactions in standard DNNs.
>
> - - -
>
> Q3: "Have the authors tried to experiment with the training methods with certified robustness? The reviewer wonders whether a similar figure (like figure 3) can be produced."
>
> A3: We have followed your suggestions to conduct **a new experiment** with the training methods with certified robustness [cite 1]. Experimental results show that adversarial attacks also mainly affect high-order interactions. We used the ResNet-50 trained with [cite 1], whose robustness is certified.
>
> Given the ResNet-50, which was trained using [cite 1], we generated adversarial examples using the PGD attack on samples from the validation set of the ImageNet dataset. We followed experimental settings of the classic PGD attack (see Lines 208-216). Then, we computed multi-order interactions in normal samples and adversarial examples on the ResNet-50 pre-trained using [cite 1]. The following table reports the difference of interactions between normal samples and adversarial examples $\Delta I^{(m)}$​​, as well as the attacking utilities of multi-order interactions $\Delta J^{(m)}$​​. The results show that adversarial attacks mainly affect high-order interactions (the second row), and low-order interactions have more attacking utilities on the model output (the third row), which verifies our conclusions.
>
> | m                                                            | 0.1n       | 0.2n  | 0.3n  | 0.4n   | 0.5n  | 0.6n  | 0.7n  | 0.8n  | 0.9n      |
> | :----------------------------------------------------------- | :--------- | :---- | :---- | :----- | :---- | :---- | :---- | :---- | :-------- |
> | $\Delta I^{(m)}$​​​ in ResNet-50 trained using [cite 1]         | 0.003      | 0.002 | 0.000 | 0.000  | 0.002 | 0.003 | 0.005 | 0.007 | **0.009** |
> | $\Delta J^{(m)} (\times 10^{-6})$​​​ ​ in ResNet-50 trained using [cite 1] | **12.527** | 7.131 | 1.148 | -0.826 | 3.149 | 4.696 | 5.500 | 5.964 | 3.820     |
>
>
>
> [cite 1] Cohen, Jeremy, Elan Rosenfeld, and Zico Kolter. Certified adversarial robustness via randomized smoothing. International Conference on Machine Learning. PMLR, 2019.

---

> > ### Comment · Reviewer_T63V · 2021-08-19
> > **Response to the authors**
> >
> > Thank the authors for responding enthusiastically.
> >
> > 1. After reading the authors' response, the reviewer still argues that the paper seems to bring limited technical novelty to the community.
> > First, the authors reuse the techniques to prove the decomposition of the network output in this paper.
> > Second, due to pretty much the same formulation, the proofs of linearity, nullity, commutativity, and symmetry follow straightforwardly. I
> > Third, according to the definition of $\Delta v(i,j,s) = v(S\cup\{i,j\}) - v(S\cup\{i\}) - v(S\cup\{j\})+ v(S))$, the equivalence between the multi-order interaction and mutual information seems straightforward as well.
> >
> > 2.Thank the authors for conducting the additional experiment on certified robustness. Much appreciated it. By comparing the scale of $\Delta I{(m}$ in Fig. 3 and the above table, such difference seems statistically indistinguishable. Also monotonicity is not observed in the third row.
> >
> > 3.The reviewer take a second time, reading this manuscript, but still fails to understand how to compute the proposed measure with seemingly combinatorial complexity.

---

> > > ### Author Response · Authors · 2021-08-19
> > > **Responses to further questions of Reviewer (T63V) in the second round**
> > >
> > > Thank you very much for your further comments, and we will answer all your concerns.
> > >
> > > Q1: "..., the reviewer still argues that the paper seems to bring limited technical novelty to the community." Proofs for properties of the interaction metric seem straightforward.
> > >
> > > A1: Beyond the answer for the same issue in the first-round rebuttal, we would like to further explain the main contribution of this paper. The contribution of this paper is **NOT** to prove the properties of the interaction metric (which is simple). Instead, our contribution is that we are the first time to **prove the relationship between the game-theoretic interaction and adversarial robustness**, to the best of our knowledge. To this end, we discover that adversarial attacks mainly affect high-order interactions, and explain the reason why adversarial training improves the adversarial robustness of DNNs. More crucially, we provide a unified theory to understand the essential mechanism shared by four defense methods [55, 26, 10], *i.e.*, these methods either discover or remove the highest-order interactions.
> > >
> > > If there are still any concerns, we welcome your further comments and expect more discussions.
> > >
> > > - - -
> > >
> > > Q2.1: "By comparing the scale of $\Delta I^{(m)}$​​​ in Fig. 3 and the above table, such difference seems statistically indistinguishable."
> > >
> > > A2.1: The answer to this question is twofold. **First, the absolute magnitude (scale) of $\Delta I^{(m)}$​​​​​ is meaningless in our analysis. Second, using the normalized interaction metric, we can strongly verify our conclusion.**
> > >
> > > First, the small magnitude of $\Delta I^{(m)}$​ in the table can be explained by the decomposition in Eq. (3):
> > >
> > > $$
> > > \Delta v(N|x)=v(N|x)-v(N|x^{\text{adv}})=\underbrace{\Delta v(\emptyset|x)+\sum_{i\in N}\Delta \phi^{(0)}(i|x)}_{\text{usually can be ignored}}+\sum _{i\ne j\in N} \sum _{m=0}^{n-2} \frac{n-1-m}{n(n-1)}\Delta I_\{ij\}^{(m)}
> > > $$
> > >
> > > the magnitude (scale) of $\Delta I^{(m)}$​​​​ is determined by the overall utility of adversarial perturbations on the network output $\Delta v(N|x)$​​​​. The ResNet-50 in the table is a certified robust model, so its $\Delta v(N|x)$​​​​ value is much smaller than the model in Figure 3. Thus, the metric $\Delta I^{(m)}$ in the table is smaller than that in Figure 3.
> > >
> > > | m                                                            | 0.1n  | 0.2n  | 0.3n  | 0.4n   | 0.5n  | 0.6n  | 0.7n  | 0.8n  | 0.9n      |
> > > | :----------------------------------------------------------- | :---- | :---- | :---- | :----- | :---- | :---- | :---- | :---- | :-------- |
> > > | $\Delta I^{(m)}$​​​ in ResNet-50 trained using [cite 1]         | 0.003 | 0.002 | 0.000 | 0.000  | 0.002 | 0.003 | 0.005 | 0.007 | **0.009** |
> > > | The normalized metric $\frac{\|\Delta I^{(m)}\|}{\sum_{m^{\prime}}\|\Delta I^{(m^{\prime})}\|}$​in ResNet-50 trained using [cite 1] | 0.097 | 0.064 | 0.012 | -0.009 | 0.064 | 0.097 | 0.161 | 0.225 | **0.289** |
> > >
> > > Second, the normalized metric $\frac{\|\Delta I^{(m)}\|}{\sum_{m^{\prime}}\|\Delta I^{(m^{\prime})}\|}$​​ is more convincing than $\Delta I^{(m)}$​. High-order ($0.9n$​-order) interaction metrics $\frac{\|\Delta I^{(m)}\|}{\sum_{m^{\prime}}\|\Delta I^{(m^{\prime})}\|}$​ are **3~4.5 times** of low-order ($0.1n$​-order, $0.2n$​-order) interaction metrics $\frac{\|\Delta I^{(m)}\|}{\sum_{m^{\prime}}\|\Delta I^{(m^{\prime})}\|}$​​, which is significant enough to verify our conclusion.
> > >
> > >
> > >
> > > Q2.2: "Also monotonicity is not observed in the third row."
> > >
> > > A2.2:  Thank you for your comments, but our conclusion is **NOT** that the attacking utility $\Delta J^{(m)}$​ monotonically decreases. Instead, we conclude that low-order interactions usually have more attacking utilities $\Delta J^{(m)}$ than high-order interactions in adversarially trained DNNs. The non-monotonicity of $\Delta J^{(m)}$​ may be caused by the data noises, but it does not hurt the conclusion.
> > >
> > > | m                                                            | 0.1n       | 0.2n  | 0.3n  | 0.4n   | 0.5n  | 0.6n  | 0.7n  | 0.8n  | 0.9n  |
> > > | :----------------------------------------------------------- | :--------- | :---- | :---- | :----- | :---- | :---- | :---- | :---- | :---- |
> > > | $\Delta J^{(m)} (\times 10^{-6})$​​​ ​ in ResNet-50 trained using [cite 1] | **12.527** | 7.131 | 1.148 | -0.826 | 3.149 | 4.696 | 5.500 | 5.964 | 3.820 |
> > >
> > > In the above table,  $\Delta J^{(m)}$ of low-order ($0.1n$-order) interactions are **2.1~3.3 times** of $\Delta J^{(m)}$ of high-order ($0.8n$-order, $0.9n$-order)  interactions, which is significant enough to verify our conclusion.
> > >
> > > If there are still any concerns, we welcome your further comments.
> > >
> > >
> > >
> > > [cite 1] Cohen, Jeremy, Elan Rosenfeld, and Zico Kolter. Certified adversarial robustness via randomized smoothing. International Conference on Machine Learning. PMLR, 2019.
> > >
> > > - - -
> > >
> > > Q3: **The new concern, computational details of the metric:** "... still fails to understand how to compute the proposed measure with seemingly combinatorial complexity."
> > >
> > > A3: We have introduced computational details of the proposed measure in Lines 203-207. We would like to clarify it again. To reduce the computational complexity, we applied the sampling-based approximation method in [58], and computed interactions at the grid level. More specifically, we split the input image into $16\times 16$ grids, and took each grid as an input variable. Then, we apply the sampling-based method [58] to approximate the $m$​-order interaction between two grids. We randomly sampled 80 pairs of grids, and for each pair of grids, we sampled 500 times to approximate the interaction.
> > >
> > > If there is still something unclear, you are welcome to post new comments, so that we can further discuss it.
> > >
> > > ---
> > >
> > > Do you think we have successfully explained all your concerns? If not, we welcome your further comments and expect more discussions. We will clarify all the discussed issues in the paper. Anyway, we believe the submission quality improves, rather than decreases, after the rebuttal.

---

> > > ### Author Response · Authors · 2021-08-31
> > > **Inquiry about any further questions**
> > >
> > > Dear Reviewer (T63V),
> > >
> > > We really appreciate your efforts during the review process. Do you have any further questions about this paper? If there are still some concerns, we welcome your further comments and expect more discussions.
> > >
> > > Thank you very much!
> > >
> > > With best regards,
> > >
> > > Paper2074 Authors

---

> ### Author Response · Authors · 2021-08-10
> **Response to Review (T63V) (Part 2)**
>
> Q4: "why can the trend of $\hat{v}(S)$ be reflected by $v(S)$?"
>
> A4: The two terms are not the same, but we can prove that there is a close relationship between $v(S)$​​​​​ and $\hat{v}(S)$​​​​​​​ when the model prediction is correct and confident. We can prove this claim in both theory and experiments.
>
> For the theoretical proof, we can prove this claim by the following deviation.
> $$
> \hat{v}(S)=H(Y|X_S)=E_{p(x_S)}H(Y|X_S=x_S)=-E_{p(x_S)}\sum_{y}p(y|x_S)\log p(y|x_S)
> $$
>
> $$
> =E_{p(x_S)}\left[-p(y=y^{\text{truth}}|x_S) \underbrace{\log p(y=y^{\text{truth}}|x_S)}_{v(S)}-\sum _{y\ne y^{\text{truth}}}p(y|x_S)\log p(y|x_S)\right]
> $$
>
> $$
> =E_{p(x_S)}\left[-\exp(v(S))v(S)-(1-p(y^{\text{truth}}|x_S))\sum_{y\ne y^{\text{truth}}}\frac{p(y|x_S)}{1-p(y^{\text{truth}}|x_S)}\log p(y|x_S)\right]
> $$
>
> $$
> =E_{p(x_S)}\left[-\exp(v(S))v(S)-(1-p(y^{\text{truth}}|x_S))\sum_{y\ne y^{\text{truth}}}\frac{p(y|x_S)}{1-p(y^{\text{truth}}|x_S)}\log \frac{p(y|x_S)}{1-p(y^{\text{truth}}|x_S)}
> +(1-p(y^{\text{truth}}|x_S))\sum_{y\ne y^{\text{truth}}}\frac{p(y|x_S)}{1-p(y^{\text{truth}}|x_S)}\log (1-p(y^{\text{truth}}|x_S))\right]
> $$
>
> $$
> = E_{p(x_S)}\left[-\exp(v(S))v(S)-(1-p(y^{\text{truth}}|x_S))H(Y^\prime|X_S=x_S)+(1-p(y^{\text{truth}}|x_S))\log (1-p(y^{\text{truth}}|x_S))\right]
> $$
>
> where $H(Y^\prime|X_S=x_S)$​​​​ denotes the entropy of the classification probability when we only consider the $C-1$​​​ incorrect categories $Y^\prime (y^{\text{truth}}\notin Y^\prime)$​​​. $\forall y\ne y^{\text{truth}}$​​​, $p^\prime(y|x_S)=\frac{p(y|x_S)}{1-p(y^{\text{truth}}|x_S)}$​​​ and $\sum_{y\ne y^{\text{truth}}} p^\prime(y|x_S)=1.$​
>
> Let $w(S)=1-p(y^{\text{truth}}|x_S)=1-\exp(v(S))$​​, which is strongly and negatively correlated to $v(S)$​​​. Then, we have
> $$
> \begin{aligned}
> \hat{v}(S)&=E_{p(x_S)}\left[-\exp(v(S))v(S)-w(S)H(Y^\prime|X_S=x_S)+w(S)\log w(S)\right]\\
> \end{aligned}
> $$
> In the above equation, both the first term $-\exp(v(S))v(S)$​​ and the third term $w(S)\log w(S)$​​ are negatively related to $v(S)$​​. For the second term, when the model prediction is correct and confident, then entropy inside incorrect categories $H(Y^\prime|X_S=x_S)$​​ will be relatively stable. Thus, $v(S)$​​ can reflect the negative trend of $\hat{v}(S)$​​.
>
> In this way, when we use $\hat{v}(S)$, we can still obtain similar conclusions that adversarial attacks mainly affect high-order interactions.
>
> Besides, we have also conducted experiments to verify the claim. Please see Figure 6 of the supplementary material. In this experiment, we used $\hat{v}(S)$​​ to compute interactions and proved that adversarial attacks mainly affect high-order interactions.
>
> We will follow your suggestions to add more discussions in the paper.
>
> ---
>
> Q5: "What are the motivations of the two equations in Eq. (4)?"
>
> A5: The motivation of Eq. (4) is to measure the discrimination power of interactions of a specific order, as discussed in Lines 267-277 of the paper. According to the efficiency property of the multi-order interactions in Line 165, the model output can be decomposed into the weighted sum of massive interaction components $\Delta v(i,j,S|x)$:
> $$
> v(N|x)=v(\emptyset|x)+\sum_{i\in N}\phi^{(0)}(i|x)+\sum_{i,j\in N, i\ne j}\sum_{m=0}^{n-2}\frac{n-1-m}{n(n-1)}E_{S\subseteq N\setminus\\{i,j\\},|S|=m}[\Delta v(i,j,S)]
> $$
> In this way, if interaction components of a certain order $m$​​​​​​ are all positive (or negative) and do not conflict with each other, it indicates that these components jointly promote or suppress the model output, showing a strong discrimination power. Otherwise, if some interaction components are positive and others are negative, their effects on the model output will be eliminated. In this case, the discrimination power of interactions is poor.
>
> In this way, we design the metric $D^{(m)}$​ in Eq. (4) to model the above phenomenon. The physical meaning of this metric is shown in the following equation.
>
> $$D^{(m)}=E_{x\in \Omega}E_{i,j\in N, i\ne j} \frac{\overbrace{|E_{S\subseteq N\setminus\{i,j\}, |S|=m} \Delta v(i,j,S|x)|}^{\text{the strength of the average utility of all components}}}{E_{S\subseteq N\setminus\{i,j\}, |S|=m} \underbrace{|\Delta v(i,j,S|x)|}_{\text{the strength of each component}}}$$
>
> The numerator measures the strength of the average utility of all interaction components between $(i,j)$ under different contexts $S$. The dominator represents the average strength of each interaction component. If $D^{(m)}$  approximates to 1, it indicates that almost all interaction components have similar effects (either positive or negative) on the model output. If $D^{(m)}$ approximates to 0, it shows that most interaction components conflict with each other and are eliminated. Therefore, Eq. (4) measures the discrimination power of interactions.
>
> ---
>
> Q6: **About the implementation of masking:** "How do the authors implement the masking? Setting to zero or mid-gray or Gaussian noise? Will different choices affect the results?"
>
> A6: We masked variables by following settings in [1], which is cited in Line 126. In [1], the masking was implemented by setting variables to their mean values over all samples in the dataset. We will clarify this setting in the paper.
>
> Furthermore, we have followed your suggestions to conduct **a new experiment** to test the effects of the choice of masking on the results. We compared the setting of zero and the setting of mean values [1]. Experimental results show that the choice of masking does not affect the conclusion that adversarial attacks mainly affect high-order interactions.
>
> For the setting of mean values, we followed the setting in [1], which has been discussed above. For the setting of zero, during the computation of interactions, we set the masked variables to zero. In order to compare the effects of such settings on our results, for each input sample, we computed its multi-order interactions two times. In the first time, we set the masked variables to zero. In the second time, we set the masked variables to mean values.
>
> In this experiment, we aimed to verify whether or not the setting of masking affects our conclusion that adversarial attacks mainly affect high-order interactions. We used the normalized metric $\frac{|\Delta I^{(m)}|}{\sum_{m^\prime}|\Delta I^{(m^\prime)}|}$​ to quantify the significance of changes in interactions of each order. In this way,  we measured the metric $\frac{|\Delta I^{(m)}|}{\sum_{m^\prime}|\Delta I^{(m^\prime)}|}$​​​ when we used zero or mean values to implement the masking. All experimental results show that no matter whether we use zero or mean values to mask variables, we can obtain a similar conclusion that adversarial attacks mainly attack high-order interactions.
>
> | m                                                            |                                                   | 0.1n  | 0.2n  | 0.3n  | 0.4n  | 0.5n  | 0.6n  | 0.7n  | 0.8n  | 0.85n | 0.9n  | 0.95n     |
> | :----------------------------------------------------------- | :------------------------------------------------ | :---- | :---- | :---- | :---- | :---- | :---- | :---- | :---- | :---- | :---- | --------- |
> | $\frac{\|\Delta I^{(m)}\|}{\sum_{m^\prime}\|\Delta I^{(m^\prime)}\|}$​​​ on standard ResNet-18 | set the masked variables to zero                  | 0.004 | 0.002 | 0.010 | 0.013 | 0.014 | 0.022 | 0.038 | 0.080 | 0.125 | 0.207 | **0.486** |
> | $\frac{\|\Delta I^{(m)}\|}{\sum_{m^\prime}\|\Delta I^{(m^\prime)}\|}$​ on standard ResNet-18 | set the masked variables to their mean values [1] | 0.003 | 0.005 | 0.008 | 0.014 | 0.021 | 0.040 | 0.066 | 0.103 | 0.138 | 0.219 | **0.383** |
> | $\frac{\|\Delta I^{(m)}\|}{\sum_{m^\prime}\|\Delta I^{(m^\prime)}\|}$​​​ on adversarially trained ResNet-18 | set the masked variables to zero                  | 0.005 | 0.008 | 0.001 | 0.012 | 0.043 | 0.074 | 0.100 | 0.112 | 0.147 | 0.228 | **0.271** |
> | $\frac{\|\Delta I^{(m)}\|}{\sum_{m^\prime}\|\Delta I^{(m^\prime)}\|}$​ on adversarially trained ResNet-18 | set the masked variables to their mean values [1] | 0.016 | 0.023 | 0.055 | 0.082 | 0.095 | 0.100 | 0.100 | 0.108 | 0.117 | 0.144 | **0.160** |
>
> - - -

---

### Official Review · Reviewer_Chfr · 2021-07-24

**Rating:** 7
**Confidence:** 5

**Summary:**

This paper provides a new perspective to explain different adversarial attacks and adversarial defenses in a "unified" way. Specifically, the authors exploit the multi-order interactions between inputs to analyze the robustness of DNNs. Based on the multi-order interaction, this paper further investigates the representation of high-order interactions and low-order interactions and draws some interesting conclusions. Based on the observed results, this paper gives a theoretical explanation of the commonality between adversarial attacks and defenses.


**Limitations And Societal Impact:**

Almost addressed the limitations and potential negative societal impact. See my concern (2).

**Main Review:**

$\textbf{Cons}:$

(1) My biggest concern of this paper is that I can not agree with your statement about "a unified view to explain different adversarial attacks and defenses".  I am very satisfied with the perspective of the utilized "multi-order interaction" to conduct adversarial attacks and defenses. However, in this paper, the discussion only happened in the gradient-based adversarial attacking method ($i.e.$, PGD and momentum attack) and adversarial training defense.  I think that these attacking/defending strategies can not represent all of the adversarial area works ($e.g.$, optimization-based attacking method and other defending methods). And thus, causing the following concern: is this proposed theory can be extended to other related works? I have reservations about that.

(2)In Line 261, the author claimed that adversarial perturbations towards adversarially trained DNNs penalized both complex features of global collaborations and simple features of local collaborations. However, there is no evidence to say that global collaborations mean complex features and local collaborations mean simple features, at least not provided or clarified (I see the statement in Line 288 but I can not easily agree). And according to Figure 4, it can be clearly witnessed that some x-order interactions, including low-order interactions, are increased ($i.e.$, not penalized).

(3) The writing of this paper should be carefully gone through. In some paragraphs, what makes me confused is that the attacker is stated as DNN. But in my opinion, the attacker who performs attacks ($i.e.$, attack low-order interactions or high-order interactions) should be an adversarial example, not a DNN.

#post-rebuttal

I am satisfied with the author's explanation, so I am willing to raise my score and give a acceptance

**Time Spent Reviewing:**

3

---

> ### Author Response · Authors · 2021-08-10
> **Response to Reviewer (Chfr)**
>
> Thank you very much for your careful review and constructive comments. We will try our best to answer all your concerns.
>
> Q1: **Can the proposed theory be extended to other related works?**  "the discussion only happened in the gradient-based adversarial attacking method (*i.e.,*, PGD and momentum attack) and adversarial training defense." "... all of the adversarial area works (*i.e.*, optimization-based attacking method and other defending methods)."
>
> A1: Yes, the proposed theory can be extended to other adversarial attacks and defending methods. We have followed your suggestions to conduct **new experiments** on the optimization-based C&W attack [cite 1] and the randomized smoothing defending method [cite 2], in order to verify the extendability of our theory. Experimental results show that the optimization-based C&W attack also mainly affects high-order interactions, and the randomized smoothing method boosts the robustness of high-order interactions.
>
> First, based on the standard ResNet-18, we conducted the $\ell_2$​​ C&W attack on samples from the validation set of the ImageNet dataset to generate adversarial examples. Then, we computed multi-order interactions $I_{\text{nor}}^{(m)}$​​ of normal samples and multi-order interactions $I_{\text{adv}}^{(m)}$​​ of adversarial examples. The following table reports the difference of interactions $\Delta I^{(m)}=I_{\text{nor}}^{(m)}-I_{\text{adv}}^{(m)}$​​, showing that the C&W attack also mainly affects high-order interactions.
>
> | m                                     | 0.1n    | 0.2n    | 0.3n   | 0.4n   | 0.5n   | 0.6n   | 0.7n   | 0.8n   | 0.85n  | 0.9n   | 0.95n      |
> | :------------------------------------ | :------ | :------ | :----- | :----- | :----- | :----- | :----- | :----- | :----- | :----- | :--------- |
> | $\Delta I^{(m)}$​​​​ under the C&W attack | -0.0001 | -0.0001 | 0.0001 | 0.0004 | 0.0008 | 0.0015 | 0.0026 | 0.0035 | 0.0035 | 0.0049 | **0.0097** |
>
> Second, given the ResNet-50, which was trained using randomized smoothing [cite 2], we generated adversarial examples using the PGD attack on samples from the validation set of the ImageNet dataset. We followed experimental settings of the classic PGD attack (see Lines 208-216). Then, we computed multi-order interactions in normal samples and adversarial examples on the pre-trained ResNet-50. The following table reports the difference of interactions between normal samples and adversarial examples $\Delta I^{(m)}$, as well as the attacking utilities of multi-order interactions $\Delta J^{(m)}$. The results show that adversarial attacks mainly attack high-order interactions (the second row), and low-order interactions have more attacking utilities on the model output (the third row), which verifies our conclusions.
>
> | m                                                            | 0.1n       | 0.2n  | 0.3n  | 0.4n   | 0.5n  | 0.6n  | 0.7n  | 0.8n  | 0.9n      |
> | :----------------------------------------------------------- | :--------- | :---- | :---- | :----- | :---- | :---- | :---- | :---- | :-------- |
> | $\Delta I^{(m)}$​​​ in ResNet-50 trained using [cite 2]         | 0.003      | 0.002 | 0.000 | 0.000  | 0.002 | 0.003 | 0.005 | 0.007 | **0.009** |
> | $\Delta J^{(m)} (\times 10^{-6})$​​​ ​ in ResNet-50 trained using [cite 2] | **12.527** | 7.131 | 1.148 | -0.826 | 3.149 | 4.696 | 5.500 | 5.964 | 3.820     |
>
>
>
> [cite 1] Nicholas Carlini and David Wagner. Towards evaluating the robustness of neural networks. In 2017 IEEE Symposium on Security and Privacy (SP), pages 39–57. IEEE, 2017.
> [cite 2] Cohen, Jeremy, Elan Rosenfeld, and Zico Kolter. Certified adversarial robustness via randomized smoothing. International Conference on Machine Learning. PMLR, 2019.
>
> - - -
>
> Q2: "However, there is no evidence to say that global collaborations mean complex features and local collaborations mean simple features, ..."
>
> A2: In this paper, the claim of global collaborations representing complex features and local collaborations representing simple features is based on the definition of $I_{ij}^{(m)}$​​ in Eq.(2). The $m$​​-order interaction $I_{ij}^{(m)}$​​ is computed as the collaboration between $(i,j)$​​ and $m$​​ contextual variables ($m$​​ pixels) $S$​​. Thus, low-order interactions represent the collaborations between $(i,j)$​​ *w.r.t.* simple contexts of a few variables, while high-order interactions consider the complex contexts of massive variables.
>
> Furthermore, this claim can also be verified from the following two perspectives.
>
> - We have visualized regions with strong low-order and high-order interactions in Figure 1 (right) to show that low-order interactions represent simple features, and that high-order interactions represent complex features. For example, in the first image, low-order interactions mainly focus on the eye of the dog, which is a simple feature. In comparison, high-order interactions represent the whole body of the dog, which is a complex feature.
> - A similar claim has been sufficiently investigated in [cite  3]. [cite 3] has proven that low-order interactions (local collaborations) mainly reflect simple and common concepts (features), and high-order interactions (global collaborations) usually represent complex and global features.
>
> [cite 3] Xu Cheng, Chuntung Chu, Yi Zheng, Jie Ren, and Quanshi Zhang, “A Game-Theoretic Taxonomy of Visual Concepts in DNNs” in arXiv:2106.10938, 2021.
>
> - - -
>
> Q3: "And according to Figure 4, it can be clearly witnessed that some x-order interactions, including low-order interactions, are increased (*i.e.*, not penalized)."
>
> A3: This is an insightful comment, but it does not conflict with our conclusions. According to Eq. (3), the sum of attacking utilities on low-order, middle-order, and high-order interactions is a constant ($\Delta v(N|x)$​​). Therefore, the penalization of high-order interactions may also cause the increase of low-order interactions to some extent. In this way, there are two effects on low-order interactions. First, adversarial attacks penalize low-order interactions. Second, the penalization of high-order interactions also boosts low-order interactions as a side effect. In such a trade-off, low-order interactions decrease in most DNNs in Figure 4. In this way, low-order interactions are increased in a few special cases, which can be explained by the above analysis. Nevertheless, we have discussed this phenomenon in Lines 228-230 of the supplementary material.
>
> - - -
>
> Q4: **About the paper writing:** " In some paragraphs, what makes me confused is that the attacker is stated as DNN. the attacker who performs attacks ... should be an adversarial example, not a DNN.  "
>
> A4: Thank you for your comments. We will fix this problem in the revision.
>
> ---

---

### Decision · Program_Chairs · 2021-09-27

**Decision:**

Accept (Poster)

**Comment:**

This paper focuses on a new perspective to explain different adversarial attacks and adversarial defenses in a unified way. The philosophy behind sounds interesting to me, namely, exploit the multi-order interactions between inputs to analyze the robustness of DNNs. This philosophy leads to the representation of high-order interactions and low-order interactions and some interesting conclusions that I have never seen.

The clarity and novelty are marginally above the bar of NeurIPS. However, there are three key issues proposed by Reviewer T63V that should be merged in your next version. First, the authors reuse the techniques to prove the decomposition of the network output in this paper, which bring limited technical novelty to the community. Please try to explain more and revise that part. Second, due to pretty much the same formulation, the proofs of linearity, nullity, commutativity, and symmetry follow straightforwardly. Third, the equivalence between the multi-order interaction and mutual information seems straightforward as well. By the way, please revise your updated version more clear and accessible, because the reviewer still fails to understand how to compute the proposed measure with seemingly combinatorial complexity.

While all reviewers had some concerns on the significance, the authors did a particularly good job in their rebuttal. Thus, most of us have agreed to accept this paper for publication. Please include the additional explanation and experimental results (especially for Reviewer T63V) in the next version.